# Dynamic Models for Two Nonreciprocally Coupled Fields: A Microscopic Derivation for Zero, One, and Two Conservation Laws

**Kristian Blom[1,2]⋆, Uwe Thiele[2,3,4]† and Aljaž Godec[1]‡**

**1** Mathematical bioPhysics group, Max Planck Institute for Multidisciplinary Sciences, Am Faßberg 11, 37077 Göttingen, Germany
**2** Institute of Theoretical Physics, University of Münster, Wilhelm-Klemm-Strasse 9, 48149 Münster, Germany
**3** Center for Data Science and Complexity (CDSC), University of Münster, Corrensstrasse 2, 48149 Münster, Germany
**4** Center for Multiscale Theory and Computation (CMTC), University of Münster, Corrensstrasse 40, 48149 Münster, Germany

⋆ kristian.blom@uni-muenster.de , † u.thiele@uni-muenster.de , ‡ agodec@mpinat.mpg.de

## Abstract

We construct dynamic models governing two nonreciprocally coupled fields for several cases with zero, one, and two conservation laws. Starting from two microscopic non-reciprocally coupled Ising models, and using the mean-field approximation, we obtain closed-form evolution equations for the spatially resolved magnetization in each lattice. Only allowing for single spin-flip dynamics, the macroscopic equations in the thermo-dynamic limit are closely related to the nonreciprocal Allen-Cahn equations, i.e. conservation laws are absent. Likewise, only accounting for spin-exchange dynamics within each lattice, the thermodynamic limit yields equations similar to the nonreciprocal Cahn-Hilliard model, i.e. with two conservation laws. In the case of spin-exchange dynamics within and between the two lattices, we obtain two nonreciprocally coupled equations that add up to one conservation law. For each of these cases, we systematically map out the linear instabilities that can arise. Moreover, combining the different dynamics gives a large number of further models. Our results provide a microscopic foundation for a broad class of nonreciprocal field theories, establishing a direct link between non-equilibrium statistical mechanics and macroscopic continuum descriptions.

## Contents

---

# 1 Introduction

During the past decade, there has been growing interest in multicomponent systems with nonreciprocal interactions. These interactions arise in a wide range of physical and biological systems, including nonequilibrium fluids [1–6], predator-prey networks [7,8], and many-body lattice models [9–15]. On the microscopic scale, nonreciprocity reflects an effective violation of Newton's third law or a breakdown of detailed balance [16,17]. At the collective level, such interactions enable mixtures to resist coarsening and instead exhibit nontrivial spatiotemporal structures, including traveling waves and sustained oscillations [2, 18–25]. A central organizing principle in these systems is the presence or absence of conservation laws. For two

nonvariationally coupled fields, models can be classified according to whether they contain zero [14, 15, 24–29], one [30–32], or two [2, 18–21, 33–37] conservation laws.

Nonreciprocal interactions between two nonconserved order parameters have been investigated through nonreciprocal extensions of classical pattern-forming models such as the Allen-Cahn and Swift-Hohenberg equations [24, 25, 29]. By incorporating asymmetric couplings between scalar fields, nonreciprocity can lead to a variety of dynamic phases, including spiral patterns [24] and chaos [29]. Similar phenomena arise in nonreciprocal spin models [26–28], where mean-field analyses and extensive computer simulations uncover a rich phase behavior including ordered, disordered, and oscillatory phases. Specifically for the square lattice it was found through simulations that nonreciprocity can induce the nucleation of droplets, which can subsequently lead to spiral patterns [26, 28]. These results have been further analyzed beyond mean-field theory [14] and extended to three-dimensional nonreciprocal spin systems [15]. While these studies on nonreciprocal spin models focus on nonconserved dynamics, a microscopic derivation of the underlying spatially extended field theories and a direct link to the nonreciprocal Allen–Cahn or Swift–Hohenberg equations has remained elusive. Furthermore, it is natural to ask how nonreciprocal interactions give rise to distinct pattern-forming behavior in conserved systems that are governed by conservation laws and in mixed systems where conserved and nonconserved quantities interact.

A prominent example of the latter class are particular reaction-diffusion systems that describe the dynamics of reactive mixtures in the presence of conservation laws [30–32, 38–45]. These systems, often inspired by biological contexts such as the Min-protein system, exhibit robust pattern formation driven by mass redistribution and local reaction kinetics [42, 44, 45]. When similar nonvariational reaction terms are included in phase-separating systems, classical coarsening mechanisms such as Ostwald ripening can be suppressed, allowing for the formation of stable and long-lived droplets [29–32, 46].

For systems with two conserved fields, the nonreciprocal Cahn-Hilliard model, originally introduced as an extension of classical variational Cahn-Hilliard models [47, 48] to describe traveling-wave instabilities [18–20] and the suppression of coarsening [18], has become a foundational tool for studying pattern formation in mixtures with nonreciprocal couplings. This framework supports a wide variety of dynamic states, including traveling bands, oscillatory regimes, localized states, microphase separation, and defect states [2, 18, 20, 21, 37, 49]. In particular, the nonreciprocal Cahn-Hilliard model has also been identified as a universal higher-order amplitude equation that governs large-scale oscillatory and stationary as well as small-scale stationary instabilities in systems with two conservation laws [35]. Extensions incorporating thermal noise have enabled detailed investigations of time-reversal symmetry breaking and transitions between static and dynamic phases [34, 36]. In addition, numerical continuation techniques and the identification of a "spurious gradient dynamics structure" [22] have revealed complex bifurcation behavior and the coexistence of distinct stationary and oscillatory states [21, 33, 37].

While these examples collectively demonstrate the rich dynamical behavior captured by nonreciprocal field theories, they are typically introduced on phenomenological grounds, based on symmetry arguments rather than derived from microscopic principles. In this work, we address this gap by deriving several nonreciprocal field equations directly from an underlying microscopic model, namely, the nonreciprocal Ising model, that consists of two nonreciprocally coupled Ising lattices. Our derivation not only yields field equations closely related to known nonreciprocal Allen-Cahn and Cahn-Hilliard models in the appropriate limits, but also a nonreciprocal reactive Cahn-Hilliard model with one conservation law, and in extension a

number of other models with combined dynamics. This approach lays the groundwork for a systematic exploration of nonreciprocal pattern formation in mixtures with and without conservation laws.

The remainder of this article is structured as follows: In Section 2, we introduce the microscopic nonreciprocal Ising model and discuss how zero, one, and two conservation laws for the magnetization can be implemented through kinetic rules for single spin-flip and spin-exchange dynamics. In Section 3, we analyze the case without any conservation law and derive the corresponding nonreciprocal Allen-Cahn model in the thermodynamic limit. We also perform a linear stability analysis, revealing the regimes with unstable stationary and oscillatory modes, and show that Turing-type instabilities are absent at the mean-field level. Section 4 considers the case with two conservation laws, leading to a nonreciprocal Cahn-Hilliard equation. Here, the linear stability mirrors that of the Allen-Cahn case. In Section 5, we study the intermediate case, deriving a nonreciprocal reactive Cahn-Hilliard model with one conservation law. In Sect. 6, we show how the kinetic rules from the preceding sections can be combined to construct all sixteen different continuum equations that correspond to all possible combinations of the allowed microscopic moves. Finally, Section 7 summarizes our main findings and outlines future research directions.

## 2 Nonreciprocal Ising Model

### 2.1 Lattice Setup and Energetics

We consider a pair of square lattices labeled $\mu \in \{a, b\}$, each with periodic boundary conditions (see Fig. 1(a)). Every lattice contains $N = N_x N_y$ spins, where each spin can attain values $\sigma_i^\mu = \pm 1$, and $i \in \{1, \ldots, N\}$ indexes its site. Spins interact with their four nearest neighbors within the same lattice, and with the corresponding spin at the same position in the opposing lattice. The local interaction energy of spin $i$ on lattice $\mu$ is given by [14, 26]

$$E_i^\mu = -\sigma_i^\mu h_i^\mu, \tag{1}$$

where the local field $h_i^\mu$ is defined as

$$h_i^\mu \equiv H_\mu + J_\mu \sum_{\langle ij \rangle} \sigma_j^\mu + K_\mu \sigma_i^\nu, \quad \nu \neq \mu, \tag{2}$$

with $\langle ij \rangle$ indicating a sum over the nearest neighbors of spin $i$, and $\nu \neq \mu$ denotes the lattice opposing lattice $\mu$. Throughout this work, energies are expressed in units of $k_B T$, where $k_B$ is the Boltzmann constant and $T$ denotes the temperature of the external bath in which the system is immersed. The first term $H_\mu$ in Eq. (2) denotes an external magnetic field acting on lattice $\mu$. The second term describes nearest-neighbor interactions within the same lattice, with $J_\mu$ the inner-lattice coupling strength. The third term accounts for the interaction of spin $\sigma_i^\mu$ with the corresponding spin at position $i$ on the opposite lattice $\nu \neq \mu$, where $K_\mu$ is the directed interlattice coupling strength. The directed couplings consist of both reciprocal and nonreciprocal components, which correspond to the symmetric part, $(K_a + K_b)/2$, and antisymmetric part, $(K_a - K_b)/2$, of the interaction, respectively. The local interaction energy given by Eq. (1) forms the basis for defining transition rates for single spin-flip and spin-exchange dynamics, as elaborated in Sects. (3)–(5). When $K_a = K_b$ (i.e. for purely reciprocal

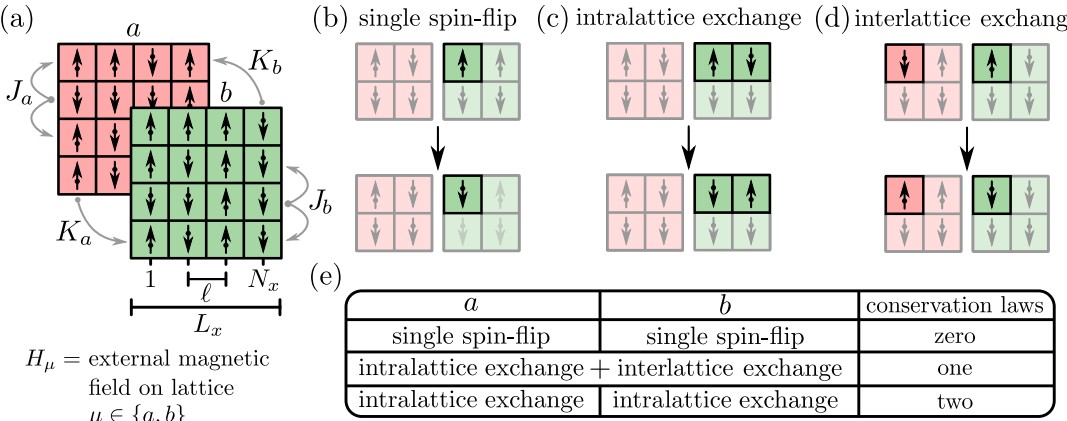

Figure 1: (a) Schematic of the lattice configuration and interaction structure in the nonreciprocal Ising model. Each square lattice has spacing $\ell$ and dimensions $\{L_x = N_x\ell, L_y = N_y\ell\}$. The parameter $J_\mu$ denotes the coupling between nearest-neighbor spins within lattice $\mu \in \{a, b\}$, $K_\mu$ represents the directed interlattice coupling, and $H_\mu$ is an external magnetic field applied to lattice $\mu$. When $K_a \neq K_b$, the interactions become nonreciprocal. (b) In single spin-flip dynamics, individual spins on each lattice flip independently. (c) In intralattice spin-exchange dynamics, two neighboring spins within the same lattice are exchanged. (d) In interlattice spin-exchange dynamics, spins at corresponding positions on opposing lattices are exchanged. (e) Summary table of the kinetic rules and their associated conservation laws for the magnetization $M^\mu(t)$ [see Eq. (3)], as discussed in Sect. 2.2

interactions), a global energy function can be defined as the sum of local energies:

$$E \equiv \frac{1}{2}\sum_\mu \sum_i E_i^\mu, \quad \text{when } K_a = K_b,$$

which corresponds to the Ising Hamiltonian for two coupled 2D lattices. For $K_a \neq K_b$, no such global energy exists, as the interlattice interaction contains a nonreciprocal component. The absence of a global energy is directly related to the absence of a Lyapunov function for the dynamics when $K_a \neq K_b$, which will be shown explicitly in Sects. (3)–(5).

## 2.2 Kinetics and Conservation Laws

We examine distinct kinetic rules that govern the dynamics of the nonreciprocal Ising model, each of which enforces a specific conservation law at the macroscopic level. These dynamics are implemented through a master equation that describes the evolution of the probability distribution $P(\boldsymbol{\sigma}; t)$ over configurations $\boldsymbol{\sigma} = \{\sigma_1^a, \sigma_1^b, \ldots, \sigma_N^a, \sigma_N^b\}$ at time $t$. Conservation laws are defined in terms of the total magnetization per lattice:

$$M^\mu(t) \equiv \sum_{\boldsymbol{\sigma}} \sum_{i=1}^N P(\boldsymbol{\sigma}; t)\sigma_i^\mu. \tag{3}$$

We consider three types of kinetic rules, each conserving a different number of total magnetizations, as summarized in Fig. 1(b)-(e) and listed below:

1. **Single spin-flip dynamics:** In this setting, also known as Glauber dynamics [50], each spin can flip independently, as illustrated in Fig. 1(b). As a result, in both lattices the

143    total magnetization is not conserved:

$$\frac{\mathrm{d}M^a(t)}{\mathrm{d}t} \neq 0, \quad \frac{\mathrm{d}M^b(t)}{\mathrm{d}t} \neq 0.$$

144    While the total magnetization can remain constant in each lattice when the system is
145    initialized in a steady state, this is not generally the case. In Sect. 3, we analyze this
146    dynamics and show that the resulting partial differential equations for the spatially re-
147    solved magnetization are, in the thermodynamic limit, closely related to the nonrecip-
148    rocally coupled Allen-Cahn equations.

149    2. **Intralattice spin-exchange dynamics:** Here, two nearest-neighbor spins within the
150    same lattice can exchange positions, as shown in Fig. 1(c). This type of spin-exchange
151    process is commonly referred to as Kawasaki dynamics [51], and conserves the total
152    magnetization in each lattice individually:

$$\frac{\mathrm{d}M^a(t)}{\mathrm{d}t} = 0, \quad \frac{\mathrm{d}M^b(t)}{\mathrm{d}t} = 0.$$

153    This scenario is analyzed in Sect. 4, and leads, in the thermodynamic limit, to two non-
154    reciprocally coupled Cahn-Hilliard equations for the spatially resolved magnetization.

155    3. **Intra- and interlattice spin-exchange dynamics:** In this case, we allow for nearest-
156    neighbor spin exchange both within and between the two lattices, as depicted in Figs. 1(c)
157    and 1(d), respectively. Consequently, the sum of the total magnetizations is conserved:

$$\frac{\mathrm{d}[M^a(t) + M^b(t)]}{\mathrm{d}t} = 0.$$

158    This kinetic rule is studied in Sect. 5, and gives rise to a nonreciprocal reactive CH model
159    with one conservation law.

160    In the following sections, we derive the macroscopic partial differential equations correspond-
161    ing to each kinetic regime by performing a mean-field and hydrodynamic coarse-graining anal-
162    ysis, followed by a linear stability assessment of uniform stationary states. Finally, in Sect. 6
163    we show how the kinetic rules in Figs. 1(b) to (d) can be combined in all possible ways to
164    construct sixteen macroscopic kinetic equations with different numbers of conservation laws.

## 2.3   Significance of the Nonreciprocal Ising Model and Conservation Laws

166    For over a century, the Ising model has been the paradigmatic framework for equilibrium phase
167    transitions [52]; analogously, the nonreciprocal Ising model can play a foundational role in
168    understanding nonequilibrium phase transitions. It offers a minimal setting for asymmetric
169    interactions between two many-body subsystems [28], with applications ranging from Ising
170    machines [53, 54] to collective opinion dynamics [55–57] and asymmetric Hopfield-type neu-
171    ral networks [58–60]. Less emphasized, however, is how the choice of dynamics (and therefore
172    the associated conservation laws) shapes the resulting phenomenology.

173        To highlight one concrete setting, consider collective opinion dynamics where the up/down
174    spins encode two opinions and the two lattices label two agent types, namely conformists and
175    contrarians [55–57]. Nonreciprocity naturally emerges in this settings: conformists prefer to
176    align with their local neighbors, whereas contrarians tend to disalign with their neighbor on

the opposing lattice and align with their neighbors within the same lattice, leading to directed cross-influences between the two groups. Different kinetic choices then probe different mechanisms: (i) *single–spin flip dynamics* models the changes of opinion under local social pressure; (ii) *intralattice spin exchange dynamics* represents spatial relocation of agents while keeping their type and opinion fixed, capturing segregation of opinions within each group; and (iii) *intra- and interlattice spin-exchange dynamics* allows swaps of agents between the two groups, enabling the segregation of opinions between the conformists and contrarians. In this way, the nonreciprocal Ising model, combined with appropriate conservation laws, provides a flexible framework to investigate how asymmetric influence and kinetic constraints govern pattern formation far from equilibrium.

Although we have illustrated these ideas with opinion dynamics, the same nonreciprocal Ising model with the appropriate kinetic constraints provides a minimal framework for other systems featuring asymmetric interactions between two subgroups [28].

## 3 Zero Conservation Laws: Single Spin-Flip Dynamics

We begin by deriving the dynamical equations associated with single spin-flip dynamics, as illustrated in Fig. 1(b). Since the total magnetization $M^{\mu}(t)$ is not conserved in either lattice, one could expect that the dynamics for the spatially resolved magnetization will be governed by a pair of nonreciprocally coupled Allen-Cahn equations [24]. Let $P(\boldsymbol{\sigma}; t)$ denote the probability of finding the system in state $\boldsymbol{\sigma} = \{\sigma_1^a, \sigma_1^b, \ldots, \sigma_N^a, \sigma_N^b\}$ at time $t$. This probability evolves according to the master equation

$$\frac{\mathrm{d}P(\boldsymbol{\sigma}; t)}{\mathrm{d}t} = \sum_{\mu} \sum_i \left[ w(-\sigma_i^{\mu}) P(\boldsymbol{\sigma}_i^{\mu}; t) - w(\sigma_i^{\mu}) P(\boldsymbol{\sigma}; t) \right], \tag{4}$$

where $\boldsymbol{\sigma}_i^{\mu} = \{\sigma_1^a, \sigma_1^b, \ldots, -\sigma_i^{\mu}, \ldots, \sigma_N^a, \sigma_N^b\}$ is the configuration obtained from configuration $\boldsymbol{\sigma}$ by flipping spin $\sigma_i^{\mu}$. The configuration space comprises $2^{2N}$ possible states. The transition rate $w(\sigma_i^{\mu})$ for flipping a spin $\sigma_i^{\mu} \to -\sigma_i^{\mu}$ is constrained by three conditions: interactions are limited to nearest neighbors, the rates attain the same functional form for each spin, and detailed balance holds when $K_a = K_b$. These assumptions yield the general Glauber-type rate [15, 27, 50]:

$$w(\sigma_i^{\mu}) = \frac{1}{2\tau} \left[ 1 - \tanh\left( \frac{\Delta E_i^{\mu}}{2} \right) \right], \tag{5}$$

where $\Delta E_i^{\mu}$ is the energy change on lattice $\mu \in \{a, b\}$ due to flipping $\sigma_i^{\mu}$, and $\tau \geq 0$ denotes the characteristic time scale of single spin-flip attempts. Using Eqs. (1) and (2), the local energy change after a single spin-flip reads

$$\Delta E_i^{\mu} = -2E_i^{\mu} = 2\sigma_i^{\mu} h_i^{\mu}.$$

From Eqs. (4) and (5), the time evolution for the expectation value of a single spin,

$$m_i^{\mu}(t) \equiv \langle \sigma_i^{\mu} \rangle(t) = \sum_{\boldsymbol{\sigma}} P(\boldsymbol{\sigma}; t) \sigma_i^{\mu}, \tag{6}$$

can be obtained directly and reads [50]

$$\tau \frac{\mathrm{d}m_i^{\mu}(t)}{\mathrm{d}t} = \langle \tanh(h_i^{\mu}) \rangle(t) - m_i^{\mu}(t). \tag{7}$$

208  This equation is not closed, as the first term on the right involves a nonlinear average over the
209  local field $h_i^\mu$ (see Eq. (2)). To address this closure problem, we apply the mean-field (MF)
210  approximation introduced in [61], enabling us to express the dynamics in a closed form.

## 3.1  Mean-Field Approximation

212  Within the MF approximation, each spin is assumed to interact with the average local field:

$$h_i^\mu \overset{\text{MF}}{=} \langle h_i^\mu \rangle \overset{(2)}{=} H_\mu + J_\mu \sum_{\langle ij \rangle} m_j^\mu + K_\mu m_i^\nu, \quad \nu \neq \mu. \tag{8}$$

213  This assumption would be exact in the case of all-to-all coupling, but for the square lattice con-
214  sidered here, it serves only as a crude first-order approximation. Inserting this approximation
215  into Eq. (7) gives

$$\tau \frac{\mathrm{d} m_i^\mu(t)}{\mathrm{d}t} = \tanh(\langle h_i^\mu \rangle(t)) - m_i^\mu(t). \tag{9}$$

216  This intermediate result has also been derived in [14, 26, 28]. Although this equation is now
217  closed, it is not yet close to the standard Allen-Cahn form. We therefore employ the identity

$$c \tanh(x) - y = -(c - y \tanh(x)) \tanh(\operatorname{arctanh}(y/c) - x), \tag{10}$$

218  for $c = 1$ together with Eq. (2), to rewrite Eq. (9) as (suppressing the time argument $t$):

$$\begin{aligned}
\tau \frac{\mathrm{d} m_i^a}{\mathrm{d}t} &= -\mathcal{M}_i^a(\mathbf{m}^a, \mathbf{m}^b) \tanh\left( \frac{\partial \mathcal{F}(\mathbf{m}^a, \mathbf{m}^b)}{\partial m_i^a} - \frac{K_a - K_b}{2} m_i^b \right), \\
\tau \frac{\mathrm{d} m_i^b}{\mathrm{d}t} &= -\mathcal{M}_i^b(\mathbf{m}^a, \mathbf{m}^b) \tanh\left( \frac{\partial \mathcal{F}(\mathbf{m}^a, \mathbf{m}^b)}{\partial m_i^b} + \frac{K_a - K_b}{2} m_i^a \right),
\end{aligned} \tag{11}$$

219  where $\mathbf{m}^\mu = \{m_1^\mu, ..., m_N^\mu\}$. Here, $\mathcal{F}(\mathbf{m}^a, \mathbf{m}^b)$ is the MF free energy defined as

$$\mathcal{F}(\mathbf{m}^a, \mathbf{m}^b) \equiv \frac{1}{2} \sum_\mu \sum_{i=1}^N \left[ \Phi(m_i^\mu) - 2H_\mu m_i^\mu - J_\mu m_i^\mu \sum_{\langle ij \rangle} m_j^\mu - K_\mu m_i^a m_i^b \right], \tag{12}$$

220  with entropy function

$$\Phi(x) \equiv (1 + x) \ln(1 + x) + (1 - x) \ln(1 - x). \tag{13}$$

221  Finally, the nonnegative mobility $\mathcal{M}_i^\mu \geq 0$ is given by

$$\mathcal{M}_i^\mu(\mathbf{m}^a, \mathbf{m}^b) \equiv 1 - m_i^\mu \tanh(\langle h_i^\mu \rangle). \tag{14}$$

222  For $N$ spins Eqs. (11) comprise a set of $2N$ nonlinearly coupled ordinary differential equations
223  that can be solved numerically upon specifying the initial conditions.

## 3.2  Lyapunov Function for Reciprocal Interactions

225  Before taking the thermodynamic limit of Eqs. (11), we verify that the dynamics reduces to
226  a gradient descent form in the reciprocal case $K_a = K_b$. In this case, the MF free energy

$\mathcal{F}(\mathbf{m}^a, \mathbf{m}^b)$ in Eq. (12) serves as a Lyapunov function of Eqs. (11). To see this, we compute its time derivative and obtain:

$$\tau \frac{d\mathcal{F}}{dt} = \tau \sum_\mu \sum_{i=1}^N \frac{\partial \mathcal{F}}{\partial m_i^\mu} \frac{dm_i^\mu}{dt} \overset{(11)}{=} -\sum_\mu \sum_{i=1}^N \mathcal{M}_i^\mu \frac{\partial \mathcal{F}}{\partial m_i^\mu} \tanh\left( \frac{\partial \mathcal{F}}{\partial m_i^\mu} \right) \leq 0,$$

where the inequality follows from the nonnegative mobilities, and from $x \tanh(x) \geq 0$ for all $x \in \mathbb{R}$. More generally, the hyperbolic tangent function in Eqs. (11) may be replaced by any odd, monotonically increasing function $f(x)$ satisfying $f(-x) = -f(x)$, such as $\sinh(x)$, without affecting the gradient descent structure. In the reciprocal case $K_a = K_b$, the resulting dynamics still ensures that $\mathcal{F}(\mathbf{m}^a, \mathbf{m}^b)$ decreases monotonically over time.

### 3.3  Thermodynamic Limit

Next, we take the thermodynamic limit of Eqs. (11). Let $\ell$ be the distance between nearest-neighbor spins as shown in Fig. 1(a), so that the total length of each side of the square lattice is given by

$$\{L_x, L_y\} = \{N_x \ell, N_y \ell\}.$$

In the thermodynamic limit, we take $\{N_x, N_y\} \to \infty$ while keeping $\{L_x, L_y\}$ fixed, and therefore the lattice spacing goes to $\ell \to 0$. In this limit, we can define a smooth continuum field $m^\mu(\mathbf{x}, t)$ as a local average over a two-dimensional box $\Lambda_{n\ell}(i)$ of linear size $n\ell$ with $n \in \mathbb{N}$, centered at site $i$ corresponding to spatial location $\mathbf{x} \in \mathbb{R}^2$ expressed in units of the lattice spacing $\ell$. Let $|\Lambda_{n\ell}(i)|$ be the number of spins in the box, then the continuum field is defined as

$$m^\mu(\mathbf{x}, t) \equiv \overset{\{L_x, L_y\}=\text{const.}}{\underset{\{N_x, N_y\} \to \infty}{\lim}} \frac{1}{|\Lambda_{n\ell}(i)|} \sum_{j \in \Lambda_{n\ell}(i)} m_j^\mu(t).$$

Given that the box size is sufficiently larger than the lattice spacing (i.e., $n \gg 1$), the continuum field $m^\mu(\mathbf{x}, t)$ becomes smooth. Therefore, finite differences can be approximated by differential operators [61]. To implement this, let $\hat{\mathbf{e}}_x = (1, 0)^\mathsf{T}$ and $\hat{\mathbf{e}}_y = (0, 1)^\mathsf{T}$ denote the shift vectors in the $x$ and $y$ directions (in units of $\ell$), respectively. In the thermodynamic limit where $\ell \to 0$ we can write the shifted fields as a Taylor expansion

$$m^\mu(\mathbf{x} \pm \hat{\mathbf{e}}_x, t) = \sum_{k=0}^\infty \frac{(\pm \partial_x)^k}{k!} m^\mu(\mathbf{x}, t) = e^{\pm \partial_x} m^\mu(\mathbf{x}, t), \tag{15}$$

$$m^\mu(\mathbf{x} \pm \hat{\mathbf{e}}_y, t) = \sum_{k=0}^\infty \frac{(\pm \partial_y)^k}{k!} m^\mu(\mathbf{x}, t) = e^{\pm \partial_y} m^\mu(\mathbf{x}, t), \tag{16}$$

where the second equality follows from the definition of the Taylor series of the exponential. Using the Taylor series gives the following gradient expansion for the sum of nearest neighbors

$$\overset{\{L_x, L_y\}=\text{const.}}{\underset{\{N_x, N_y\} \to \infty}{\lim}} \sum_{\langle ij \rangle} m_j^\mu(t) = 2[\cosh(\partial_x) + \cosh(\partial_y)] m^\mu(\mathbf{x}, t) = 4m^\mu(\mathbf{x}, t) + \nabla^2 m^\mu(\mathbf{x}, t) + \mathcal{O}(\nabla^4 m^\mu), \tag{17}$$

where $\nabla^2$ is the Laplace operator. Applying this limit to the free energy (12) and mobility (14), we obtain from Eqs. (11) the partial differential equations (omitting the arguments $(\mathbf{x}, t)$)

$$\tau \frac{\partial m^a}{\partial t} = -\mathcal{M}^a(m^a, m^b) \tanh\left( \frac{\delta \mathcal{F}[m^a, m^b]}{\delta m^a} - \frac{K_a - K_b}{2} m^b \right),$$

$$\tau \frac{\partial m^b}{\partial t} = -\mathcal{M}^b(m^a, m^b) \tanh\left( \frac{\delta \mathcal{F}[m^a, m^b]}{\delta m^b} + \frac{K_a - K_b}{2} m^a \right). \tag{18}$$

The free-energy functional $\mathcal{F}[m^a, m^b]$ is the thermodynamic limit of Eq. (12), which reads

$$\mathcal{F}[m^a, m^b] \equiv \frac{1}{2} \sum_\mu \int d\mathbf{x} \left[ \Phi(m^\mu) - 2H_\mu m^\mu - J_\mu \left( 4(m^\mu)^2 + m^\mu \nabla^2 m^\mu \right) - K_\mu m^a m^b \right]$$

$$\stackrel{\text{p.i.}}{=} \frac{1}{2} \int d\mathbf{x} \left[ f(m^a, m^b) + \sum_\mu J_\mu |\nabla m^\mu|^2 \right]. \tag{19}$$

From the first to the second line we applied partial integration to the gradient term while assuming vanishing Neumann boundary conditions, and we identified the local MF free-energy density:

$$f(m^a, m^b) \equiv \sum_\mu \left[ \Phi(m^\mu) - 2H_\mu m^\mu - 4J_\mu (m^\mu)^2 - K_\mu m^a m^b \right]. \tag{20}$$

The mobilities appearing in Eqs. (18) are the thermodynamic limit of Eq. (14), and read

$$\mathcal{M}^\mu(m^a, m^b) \equiv 1 - m^\mu \tanh\left( \operatorname{arctanh}(m^\mu) - \left[ \frac{\delta \mathcal{F}[m^a, m^b]}{\delta m^\mu} + (-1)^{\delta_{\mu,a}} \frac{K_a - K_b}{2} m^\nu \right] \right), \quad \nu \neq \mu,$$

$$= 1 - m^\mu \tanh\left( H_\mu + J_\mu \left[ 4 + \nabla^2 \right] m^\mu + K_\mu m^\nu \right), \quad \nu \neq \mu. \tag{21}$$

Note that the first line in Eq. (21) follows directly from Eq. (10) and will be used in the next section to perform a Taylor expansion.

## 3.4 Expansion Close To Stationary States

To clarify the connection between Eqs. (18) and the nonreciprocal Allen-Cahn equations [24], note that in the vicinity of stationary states the argument of the hyperbolic tangent is small. For brevity we define

$$x^\mu(m^a, m^b) \equiv \frac{\delta \mathcal{F}[m^a, m^b]}{\delta m^\mu} + (-1)^{\delta_{\mu,a}} \frac{K_a - K_b}{2} m^\nu, \quad \nu \neq \mu,$$

such that $|x^\mu| \ll 1$ close to stationarity. The hyperbolic tangent can then be expanded as

$$\tanh(x^\mu) = x^\mu + \mathcal{O}((x^\mu)^3).$$

At the same time, we expand the mobility in powers of $x^\mu$ using the first line in Eq. (21), resulting in

$$\mathcal{M}^\mu(m^a, m^b) = 1 - m^\mu \tanh(\operatorname{arctanh}(m^\mu) - x^\mu) = 1 - (m^\mu)^2 + \mathcal{O}(x^\mu).$$

Consequently, close to stationarity, Eqs. (18) reduce (to linear order in $x^\mu$) to the nonreciprocal Allen-Cahn equations with quadratic mobilities [24]:

$$\tau \frac{\partial m^a}{\partial t} \simeq -[1 - (m^a)^2] \left( \frac{\delta \mathcal{F}[m^a, m^b]}{\delta m^a} - \frac{K_a - K_b}{2} m^b \right),$$

$$\tau \frac{\partial m^b}{\partial t} \simeq -[1 - (m^b)^2] \left( \frac{\delta \mathcal{F}[m^a, m^b]}{\delta m^b} + \frac{K_a - K_b}{2} m^a \right). \tag{22}$$

In this sense, Eqs. (18) constitute a nonlinear extension of the nonreciprocal Allen-Cahn model.

### 3.5  Linear Stability Analysis

To gain insight into the solutions of Eqs. (18), we determine the linear stability of the uniform steady state. For a nonzero magnetic field $H_\mu \neq 0$, the uniform steady state $m^\mu(\mathbf{x}, t) = m_0^\mu$ satisfies the transcendental equation

$$m_0^\mu = \tanh(H_\mu + 4 J_\mu m_0^\mu + K_\mu m_0^\nu), \quad \nu \neq \mu, \tag{23}$$

and for $H_\mu = 0$ we consider the trivial steady state $m_0^\mu = 0$, and perturb it harmonically, i.e.

$$m^\mu(\mathbf{x}, t) = m_0^\mu + \delta m^\mu \exp(i\mathbf{k} \cdot \mathbf{x} + \lambda t), \tag{24}$$

where $|\delta m^\mu| \ll 1$ are the perturbation amplitudes and $\mathbf{k} = (k_x, k_y)^\mathrm{T}$ denotes the wavevector. Substituting Eq. (24) into Eqs. (18) and linearizing in $\delta m^\mu$ yields the eigenvalue problem

$$\lambda \begin{pmatrix} \delta m^a \\ \delta m^b \end{pmatrix} = \underline{\mathbf{L}} \begin{pmatrix} \delta m^a \\ \delta m^b \end{pmatrix}, \quad \text{with} \quad \underline{\mathbf{L}} \equiv \frac{1}{\tau} \begin{pmatrix} \tilde{J}_a(4 - |\mathbf{k}|^2) - 1 & \tilde{K}_a \\ \tilde{K}_b & \tilde{J}_b(4 - |\mathbf{k}|^2) - 1 \end{pmatrix}, \tag{25}$$

where $|\mathbf{k}|^2 = k_x^2 + k_y^2$, and we have defined the rescaled coupling constants

$$\tilde{J}_\mu \equiv J_\mu[1 - (m_0^\mu)^2], \quad \tilde{K}_\mu \equiv K_\mu[1 - (m_0^\mu)^2]. \tag{26}$$

Since $m_0^\mu \in [-1, 1]$, a nonzero value of $m_0^\mu$ modifies only the magnitude — and not the sign — of the effective couplings $\tilde{J}_\mu$ and $\tilde{K}_\mu$ relative to the original couplings $J_\mu$ and $K_\mu$. From Eq. (25) we find that the eigenvalues of $\underline{\mathbf{L}}$, denoted as $\lambda_\pm$, satisfy the dispersion relation

$$\lambda_\pm = \frac{1}{2\tau} \left[ (\tilde{J}_a + \tilde{J}_b)(4 - |\mathbf{k}|^2) - 2 \pm \sqrt{4\tilde{K}_a \tilde{K}_b + (\tilde{J}_a - \tilde{J}_b)^2 (4 - |\mathbf{k}|^2)^2} \right], \tag{27}$$

that provides the growth rates $\mathrm{Re}(\lambda_\pm)$ and the frequencies $\mathrm{Im}(\lambda_\pm)$. The corresponding eigenvectors (for $\tilde{K}_b \neq 0$) are given by

$$\begin{pmatrix} \delta m_\pm^a \\ \delta m_\pm^b \end{pmatrix} = \begin{pmatrix} [\lambda_\pm + 1 - \tilde{J}_b(4 - |\mathbf{k}|^2)]/\tilde{K}_b \\ 1 \end{pmatrix}.$$

From Eq. (24), we see that the perturbation grows exponentially in time if $\mathrm{Re}(\lambda) > 0$, indicating a linear instability. Moreover, if $\mathrm{Im}(\lambda) \neq 0$, the instability also exhibits temporal oscillations. In the following sections, we determine the parameter regimes under which these instabilities occur.

### 3.5.1  Hopf Instability

A Hopf instability (also referred to as a type-III$_\mathrm{O}$ or large-scale oscillatory instability [35,62]) occurs when $\mathrm{Re}(\lambda_\pm) = 0$ and $\mathrm{Im}(\lambda_\pm) \neq 0$ at zero wavenumber. To streamline the notation, we introduce the following auxiliary variables, which help characterize the parameter space where Hopf (and other) instabilities arise:

$$\mathcal{K}_1(\tilde{J}_a, \tilde{J}_b) \equiv -4(\tilde{J}_a - \tilde{J}_b)^2, \tag{28}$$

$$\mathcal{K}_2(\tilde{J}_a, \tilde{J}_b) \equiv -(\tilde{J}_a - \tilde{J}_b)^2 / (\tilde{J}_a + \tilde{J}_b)^2. \tag{29}$$

The onset of a Hopf instability occurs at $|\mathbf{k}| = 0$ when

$$\tilde{J}_a + \tilde{J}_b = 1/2, \quad \tilde{K}_a \tilde{K}_b < \mathcal{K}_1(\tilde{J}_a, \tilde{J}_b).$$

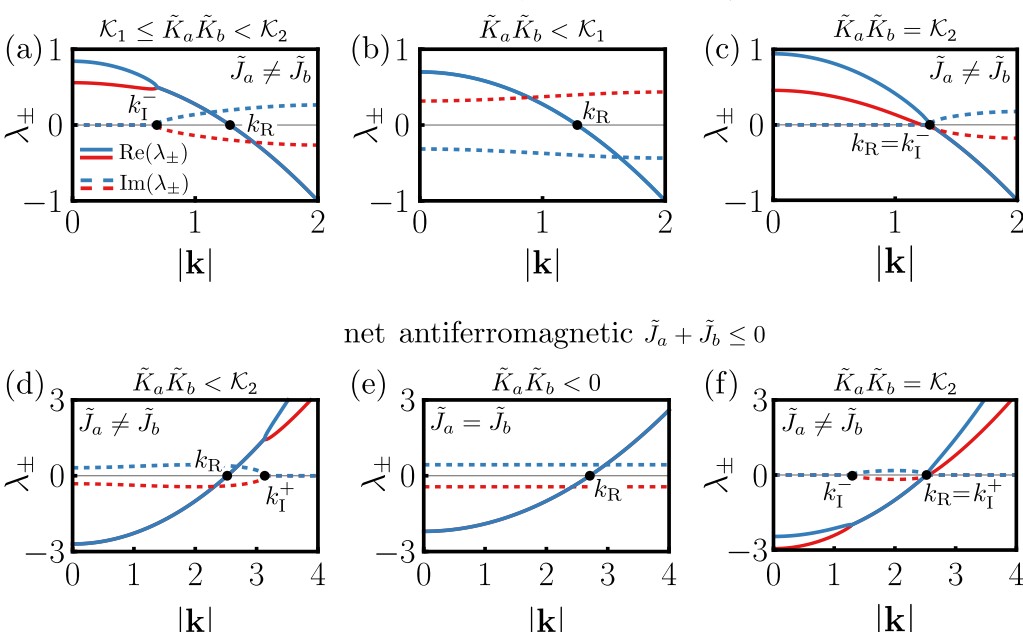

Figure 2: (a)-(f) Dispersion relations for single spin-flip dynamics [see Eq. (27)]. Panels (a)-(c) correspond to the net ferromagnetic regime with $\tilde{J}_a + \tilde{J}_b \geq 1/2$, while panels (d)-(f) correspond to the net antiferromagnetic regime with $\tilde{J}_a + \tilde{J}_b \leq 0$. In panels (a) and (d), a band of unstable stationary wavenumbers is observed, characterized by $\mathrm{Re}(\lambda_\pm) \geq 0$ and $\mathrm{Im}(\lambda_\pm) = 0$, along with an intermediate band of unstable oscillatory modes where $\mathrm{Re}(\lambda_\pm) \geq 0$ and $\mathrm{Im}(\lambda_\pm) > 0$. In panels (b) and (e), only unstable oscillatory modes are present. Panels (c) and (f) feature a critical exceptional point at $|\mathbf{k}| = k_\mathrm{R} = k_\mathrm{I}^\pm$, where $\mathrm{Re}(\lambda_\pm) = \mathrm{Im}(\lambda_\pm) = 0$. The auxiliary functions $\mathcal{K}_1(\tilde{J}_a, \tilde{J}_b)$ and $\mathcal{K}_2(\tilde{J}_a, \tilde{J}_b)$ are defined in Eqs. (28)-(29), and the wavenumbers $k_\mathrm{I}^\pm$ and $k_\mathrm{R}$ are given by Eqs. (31)-(32). Parameter values $(\tilde{J}_a, \tilde{J}_b, \tilde{K}_a\tilde{K}_b)$ used in each panel are given by: (a) $(0.5, 0.35, -0.07)$, (b) $(0.5, 0.35, -0.19)$, (c) $(0.5, 0.35, -0.0311419)$, (d) $(-0.5, -0.35, -0.14)$, (e) $(-0.3, -0.3, -0.14)$, (f) $(-0.5, -0.35, -0.0311419)$.

Since $\mathcal{K}_1(\tilde{J}_a, \tilde{J}_b) \leq 0$, a Hopf instability can only occur when $\tilde{K}_a$ and $\tilde{K}_b$ have opposite signs. This, in turn, implies that the original couplings $K_a$ and $K_b$ must also have opposite signs, which subsequently implies that the nonreciprocal interactions dominate the reciprocal ones. The perturbation modes given by Eq. (24) contain unstable oscillatory modes when $\mathrm{Re}(\lambda_\pm) \geq 0$ and $\mathrm{Im}(\lambda_\pm) \neq 0$, and unstable stationary modes when $\mathrm{Re}(\lambda_\pm) \geq 0$ and $\mathrm{Im}(\lambda_\pm) = 0$. Figure 2 shows typical dispersion relations with unstable oscillatory (and stationary modes) which arise in the following two regimes:

- In the *net ferromagnetic regime* we are above onset of the (large-scale) Hopf instability in the parameter range

$$\tilde{J}_a + \tilde{J}_b \geq 1/2, \quad \tilde{K}_a\tilde{K}_b < \mathcal{K}_2(\tilde{J}_a, \tilde{J}_b).$$

Because the eigenvalues attain the following asymptotic form for $|\mathbf{k}| \to \infty$

$$\lim_{|\mathbf{k}| \to \infty} \lambda_\pm \simeq \frac{1}{2\tau} \begin{cases} -(\tilde{J}_a + \tilde{J}_b \mp |\tilde{J}_a - \tilde{J}_b|)\,|\mathbf{k}|^2, & \tilde{J}_a \neq \tilde{J}_b, \\ -(\tilde{J}_a + \tilde{J}_b)|\mathbf{k}|^2 \pm 2\sqrt{\tilde{K}_a\tilde{K}_b}, & \tilde{J}_a = \tilde{J}_b, \end{cases} \tag{30}$$

303    we find that small-scale instabilities are suppressed in this regime, i.e.

$$\lim_{|\mathbf{k}|\to\infty} \mathrm{Re}(\lambda_\pm) \to -\infty,$$

304    as visible from Fig. 2(a)-(c).

305    In Fig. 2(a) we observe the presence of both unstable stationary and oscillatory modes,
306    which can only occur when $\tilde{J}_a \neq \tilde{J}_b$[1] and the interlattice couplings obey

$$\mathcal{K}_1(\tilde{J}_a, \tilde{J}_b) \leq \tilde{K}_a \tilde{K}_b < \mathcal{K}_2(\tilde{J}_a, \tilde{J}_b).$$

307    In this parameter regime the band of unstable oscillatory modes appear in the range

$$k_\mathrm{I}^- \leq |\mathbf{k}| \leq k_\mathrm{R},$$

308    and the band of unstable stationary modes appear in the range

$$0 \leq |\mathbf{k}| \leq k_\mathrm{I}^-,$$

309    where the lower and upper bound are defined as ($k_\mathrm{I}^+$ is relevant for the next section)

$$k_\mathrm{I}^\pm \equiv [4 \pm 2(-\tilde{K}_a\tilde{K}_b)^{1/2}/|\tilde{J}_a - \tilde{J}_b|]^{1/2} = 2[1 \pm (\tilde{K}_a\tilde{K}_b/\mathcal{K}_1)^{1/2}]^{1/2}, \quad (31)$$

$$k_\mathrm{R} \equiv [4 - 2/(\tilde{J}_a + \tilde{J}_b)]^{1/2} \qquad = 2[1 - \mathrm{sgn}(\tilde{J}_a + \tilde{J}_b)(\mathcal{K}_2/\mathcal{K}_1)^{1/2}]^{1/2}, \quad (32)$$

310    where $\mathrm{sgn}(x) = 1$ for $x > 0$ and $\mathrm{sgn}(x) = -1$ for $x < 0$. The interpretation of these
311    wavenumbers is as follows: At $|\mathbf{k}| = k_\mathrm{I}^-$, $\mathrm{Im}(\lambda_\pm)$ switches from zero to a nonzero value,
312    while at $|\mathbf{k}| = k_\mathrm{R}$ the real part vanishes, i.e., $\mathrm{Re}(\lambda_\pm) = 0$.

In Fig. 2(b) we observe that the unstable stationary modes have vanished and only the
unstable oscillatory modes remain. This occurs when $\tilde{J}_a = \tilde{J}_b$ and/or

$$\tilde{K}_a \tilde{K}_b < \mathcal{K}_1(\tilde{J}_a, \tilde{J}_b).$$

313    For these parameter values $k_\mathrm{I}^-$ becomes imaginary and therefore only the band of unsta-
314    ble oscillatory modes remain.

315    In both cases, the most unstable oscillatory mode — corresponding to the maximum of
316    $\mathrm{Re}(\lambda_\pm)$ with $\mathrm{Im}(\lambda_\pm) \neq 0$ — always occurs at the lower edge of the unstable oscillatory
317    band as seen from Fig. 2(a)-(b).

318    • In the *net antiferromagnetic regime* we have a band of oscillatory instabilities at large
319    wave numbers that occur in the parameter range

$$\tilde{J}_a + \tilde{J}_b \leq 0, \quad \tilde{K}_a \tilde{K}_b < \mathcal{K}_2(\tilde{J}_a, \tilde{J}_b),$$

320    as shown in Fig. 2(d)-(e). In this regime, the asymptotic scaling for $|\mathbf{k}| \to \infty$ given by
321    Eq. (30) shows that the smallest-scale modes are most unstable, i.e.

$$\lim_{|\mathbf{k}|\to\infty} \mathrm{Re}(\lambda_\pm) \to \infty.$$

322    The physical origin of this behavior, that one might call an "ultraviolett catastrophe"
323    lies in the antiferromagnetic coupling on the original Ising lattice. It favors alternating
324    spin orientations and thus promotes pattern formation on the scale of individual spins

---

[1]Note that $\tilde{J}_a \neq \tilde{J}_b$ does *not* imply $J_a \neq J_b$; e.g. for $J_a = J_b$, $K_a = K_b$, $H_a \neq H_b$, we have $\tilde{J}_a \neq \tilde{J}_b$ (see Eq. (26)). Similarly, $\tilde{J}_a = \tilde{J}_b$ does *not* imply $J_a = J_b$.

and results in the divergence of the eigenvalues for $|\mathbf{k}| \to \infty$. To "renormalize " this divergence, it is necessary to include higher-order terms in the gradient expansion given by Eq. (17). The resulting higher-order model is shown in Sect. 7 and is closely related to the nonreciprocally coupled Swift-Hohenberg equations (not shown) studied in [25, 29] and features small-scale stationary and oscillatory instabilities.

The band of unstable oscillatory modes emerges in the range

$$k_{\mathrm{R}} \le |\mathbf{k}| < k_{\mathrm{I}}^+, \tag{33}$$

as shown in Fig. 2(d)-(e). When $\tilde{J}_a \neq \tilde{J}_b$, the upper wavenumber $k_{\mathrm{I}}^+$ remains finite, and all unstable modes with $|\mathbf{k}| \ge k_{\mathrm{I}}^+$ are stationary as shown in Fig. 2(d). In contrast, when $\tilde{J}_a = \tilde{J}_b$, the upper wavenumber diverges, i.e. $k_{\mathrm{I}}^+ \to \infty$ since $\mathcal{K}_1 = 0$, and all unstable modes are oscillatory, as illustrated in Fig. 2(e).

In both cases, the most unstable oscillatory mode — corresponding to the maximum of $\mathrm{Re}(\lambda_\pm)$ with $\mathrm{Im}(\lambda_\pm) \neq 0$ — always occurs at the higher edge of the unstable oscillatory band as seen from Fig. 2(d)-(e).

### 3.5.2 Critical Exceptional Point

Based on the previous analysis, we can identify a special point in parameter space where the wavenumbers $k_{\mathrm{R}}$ and $k_{\mathrm{I}}^\pm$ coincide, i.e., two real eigenvalues coalesce at zero and become complex conjugate pairs. At this codimension-two point, which is very similar to a Takens-Bodganov point[2] [63, 64] and nowadays sometimes called a "critical exceptional point" [1, 2, 65], the linear stability matrix $\underline{\mathbf{L}}$ becomes nondiagonalizable and its Jordan normal form reads

$$\underline{\mathbf{L}} = \begin{pmatrix} 0 & 1 \\ 0 & 0 \end{pmatrix}.$$

Such critical points are associated with qualitative changes in the bifurcation behavior. Here, they are of codimension two as they require two constraints set by the parameter value

$$\tilde{K}_a \tilde{K}_b = \mathcal{K}_2(\tilde{J}_a, \tilde{J}_b),$$

and at specific wavenumber;

$$|\mathbf{k}| = k_{\mathrm{R}} = \begin{cases} k_{\mathrm{I}}^-, & \tilde{J}_a + \tilde{J}_b \ge 1/2, \\ k_{\mathrm{I}}^+, & \tilde{J}_a + \tilde{J}_b \le 0, \end{cases}$$

in the case that $\tilde{J}_a \neq \tilde{J}_b$. At this critical point, which is shown in Fig. 2(c) and (f), the dispersion relation undergoes a transition from regimes dominated by either purely stationary or oscillatory unstable modes to a case where the band of unstable wavenumbers contains both stationary and oscillatory modes.

### 3.5.3 Allen-Cahn Instability

An Allen-Cahn instability (also referred to as a type-III$_{\mathrm{S}}$ or large-scale stationary instability [35, 62]) arises when $\mathrm{Re}(\lambda_+) = 0$ and $\mathrm{Im}(\lambda_\pm) = 0$ at $|\mathbf{k}| = 0$. Since $\mathrm{Re}(\lambda_+) \ge \mathrm{Re}(\lambda_-)$, the onset of instability is determined by $\lambda_+$, and occurs when

$$\tilde{J}_a + \tilde{J}_b = 1/2, \quad \tilde{K}_a \tilde{K}_b \ge \mathcal{K}_1(\tilde{J}_a, \tilde{J}_b).$$

---

[2]Verification that the degeneracy is exactly a Takens–Bogdanov point requires an analysis of the nonlinear bifurcation structure.

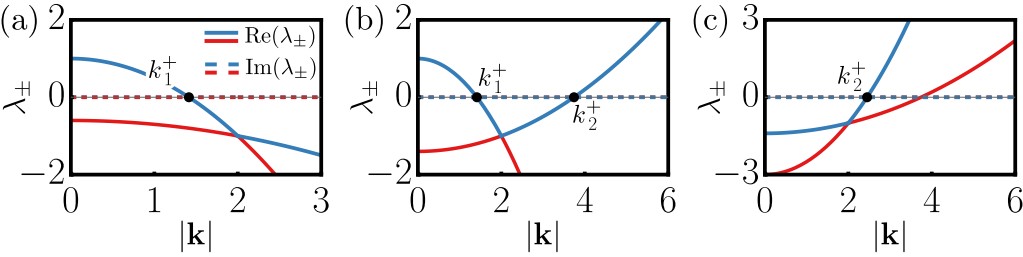

Figure 3: (a)-(c) Dispersion relations for single spin-flip dynamics [see Eq. (27)] in the presence of unstable stationary modes. Panels (b) and (c) show the emergence of a high-wavenumber band of unstable modes, resulting from effective antiferromagnetic coupling. This instability drives microphase separation, characterized by pattern formation on microscopic length scales. Parameter values $(\tilde{J}_a, \tilde{J}_b, \tilde{K}_a \tilde{K}_b)$ used in each panel are given by: (a) $(0.5, 0.1, 0)$, (b) $(0.5, -0.1, 0)$, (c) $(-0.5, -0.1, 0)$.

Above onset, unstable stationary wavenumbers with $\mathrm{Re}(\lambda_+) \geq 0$ and $\mathrm{Im}(\lambda_\pm) = 0$ can emerge within one or both of the following wavenumber intervals:

$$0 \leq |\mathbf{k}| \leq k_1^+ \quad \text{and/or} \quad |\mathbf{k}| \geq k_2^+,$$

where the wavenumbers $k_1^+$ and $k_2^+$ are given by

$$k_i^+ = \left[ 4 - \frac{1}{2} \left( \frac{1}{\tilde{J}_a} + \frac{1}{\tilde{J}_b} \right) - (-1)^i \frac{\sqrt{(\tilde{J}_a - \tilde{J}_b)^2 + 4 \tilde{J}_a \tilde{J}_b \tilde{K}_a \tilde{K}_b}}{2 \tilde{J}_a \tilde{J}_b} \right]^{1/2}, \quad i = 1, 2. \tag{34}$$

In Fig. 3, we present three types of dispersion relations that exhibit stationary instabilities. Notably, in panels (b) and (c), we observe a large-wavenumber band for $|\mathbf{k}| \geq k_2^+$, characterized by $\mathrm{Re}(\lambda_+) \geq 0$ and $\mathrm{Im}(\lambda_+) = 0$. This large-wavenumber band corresponds to very small-scale patterns and drives a process commonly referred to as microphase separation [66]. The physical mechanism underlying this behavior stems from the antiferromagnetic coupling, which promotes antiferromagnetic order and thereby induces patterns at the scale of individual spins. As in previous cases, the divergence of $\lambda_+$ as $|\mathbf{k}| \to \infty$ is unphysical and relates to an ultraviolet catastrophe. To resolve this apparent inconsistency, it is essential to incorporate higher-order terms in the gradient expansion defined in Eq. (17), which is shown in Sect. 7.

### 3.5.4 Absence of a Turing Instability

It is well established that the nonreciprocal Allen–Cahn and Cahn–Hilliard models exhibit a Turing instability [33, 37] (i.e., a type-$\mathrm{I}_S$ or small-scale stationary instability [35, 62]), corresponding to the onset of a positive growth rate at a nonzero wavenumber $|\mathbf{k}| > 0$. In contrast, we demonstrate here that the nonreciprocal Ising model, as introduced in Sec. 2, does not support such an instability on the MF level. If a Turing instability were present, it would manifest within a finite wavenumber band:

$$0 < \min(k_1^+, k_2^+) \leq |\mathbf{k}| \leq \max(k_1^+, k_2^+),$$

where the eigenvalue $\lambda_+$ vanishes at $k_{1,2}^+$. At the onset of a Turing instability, these two critical wavenumbers coincide, i.e., $k_1^+ = k_2^+ > 0$ [33], i.e. a local maximum of $\lambda_+$ touches zero. From Eq. (34), the condition $k_1^+ = k_2^+ > 0$ leads to the following inequality:

$$8 - \frac{1}{\tilde{J}_a} - \frac{1}{\tilde{J}_b} > 0, \tag{35}$$

377 along with the constraint,

$$(\tilde{J}_a - \tilde{J}_b)^2 + 4\tilde{J}_a\tilde{J}_b\tilde{K}_a\tilde{K}_b = 0. \tag{36}$$

378 To evaluate whether $\lambda_+$ can attain a local maximum at $k_{1,2}^+$, we compute its first derivative
379 and insert Eq. (36):

$$\left.\frac{\partial\lambda_+}{\partial|\mathbf{k}|}\right|_{k_{1,2}^+} = \frac{1}{\tau}\begin{cases}-(\tilde{J}_a+\tilde{J}_b)[16-2/\tilde{J}_a-2/\tilde{J}_b]^{1/2}, & \tilde{J}_a\tilde{J}_b \geq 0,\\ 0, & \tilde{J}_a\tilde{J}_b < 0.\end{cases}$$

380 Thus, for $k_{1,2}^+$ to be a stationary point of $\lambda_+$, we require either $\tilde{J}_a\tilde{J}_b < 0$, or the condition
381 $8\tilde{J}_a\tilde{J}_b - \tilde{J}_a - \tilde{J}_b = 0$ along with $\tilde{J}_a\tilde{J}_b \geq 0$. However, the latter is incompatible with the inequal-
382 ity in Eq. (35), leaving only $\tilde{J}_a\tilde{J}_b < 0$ as a valid possibility. To determine the nature of the
383 stationary point, we examine the second derivative:

$$\left.\frac{\partial^2\lambda_+}{\partial|\mathbf{k}|^2}\right|_{k_{1,2}^+} = \frac{1}{\tau}\left(\frac{\tilde{J}_a\tilde{J}_b}{\tilde{J}_a-\tilde{J}_b}\right)^2\left[8-\frac{1}{\tilde{J}_a}-\frac{1}{\tilde{J}_b}\right] > 0,$$

384 where the inequality follows directly from Eq. (35). This shows that $k_{1,2}^+$ always corresponds
385 to a local minimum of $\lambda_+$, rather than a maximum. And since a local minimum cannot belong
386 to a Turing instability, we conclude that the nonreciprocal Ising model cannot exhibit a Turing
387 instability.

388     The reason that the nonreciprocal Allen-Cahn model (or any general two-field reaction-
389 diffusion system) can exhibit a Turing instability, whereas the considered nonreciprocal Ising
390 model on the MF level cannot, lies in the structure of their respective free-energy functionals.
391 In the Ising model, the gradient energy coefficient is given by $J_\mu$, and is therefore intrinsically
392 tied to the form of the local free-energy density. Consequently, the ratio of gradient energy
393 coefficients—an essential parameter for the emergence of a Turing instability [37]—cannot
394 be tuned independently without simultaneously altering the underlying thermodynamic land-
395 scape. In contrast, for general reaction-diffusion systems, the gradient energy (or diffusion)
396 coefficients are independent transport parameters. To make this constraint explicit, we revisit
397 the previous analysis using a modified free-energy functional in which the gradient energy
398 coefficients are replaced by independent parameters $\kappa_\mu$:

$$\mathcal{F}[m^a, m^b] = \frac{1}{2}\int d\mathbf{x}\left[f(m^a, m^b) + \sum_\mu \kappa_\mu|\nabla m^\mu|^2\right],$$

399 where $f(m^a, m^b)$ is defined by Eq. (20). We define the rescaled gradient energy coefficients
400 as

$$\tilde{\kappa}_\mu \equiv \kappa_\mu[1 - (m_0^\mu)^2].$$

401 Within this framework, the onset of a Turing instability occurs when the nonreciprocal cou-
402 plings satisfy

$$\tilde{K}_a\tilde{K}_b = -\tilde{\kappa}_a\tilde{\kappa}_b\left(\frac{1-4\tilde{J}_b}{\tilde{\kappa}_b} - \frac{1-4\tilde{J}_a}{\tilde{\kappa}_a}\right)^2,$$

403 provided that the following conditions are met:

$$0 < \tilde{\kappa}_a < \tilde{\kappa}_b, \quad \tilde{\kappa}_a(1-4\tilde{J}_b) \geq \tilde{\kappa}_b(1-4\tilde{J}_a), \quad \tilde{\kappa}_a(1-4\tilde{J}_b) \geq -\tilde{\kappa}_b(1-4\tilde{J}_a).$$

404 These requirements cannot be satisfied simultaneously when $\tilde{\kappa}_a = \tilde{J}_a$ and $\tilde{\kappa}_b = \tilde{J}_b$. In that
405 case, the second condition would imply

$$\tilde{J}_a(1-4\tilde{J}_b) \geq \tilde{J}_b(1-4\tilde{J}_a) \quad \Rightarrow \quad \tilde{J}_a \geq \tilde{J}_b,$$

which contradicts the first condition $0 < \tilde{J}_a < \tilde{J}_b$. This incompatibility confirms that the necessary conditions for a Turing instability cannot be met within the nonreciprocal Ising model on the MF level. It remains an open question whether other approximation techniques such as the pair approximation [14] can lead to a Turing instability in the nonreciprocal Ising model.

If higher-order gradient terms were incorporated into the MF free-energy functional (19) to regularize the unphysical antiferromagnetic divergence of the eigenvalues (see Fig. 2(d)-(f) and Fig. 3(b)-(c)), the resulting nonreciprocal Swift-Hohenberg-type models system would inherently support small-scale stationary (Turing) and oscillatory (wave) instabilities, cf. [25, 29].

# 4   Two Conservation Laws: Intralattice Exchange Dynamics

Next, we focus on intralattice spin-exchange dynamics, as illustrated in Fig. 1(c). Since the total magnetization in each of the two lattices is conserved, we expect that the dynamics of the spatially resolved magnetization $m^\mu(\mathbf{x}, t)$ is governed by a pair of nonreciprocally coupled Cahn-Hilliard equations [2, 18–21, 33–35]. The probability $P(\boldsymbol{\sigma}; t)$ of finding the system in configuration $\boldsymbol{\sigma}$ at time $t$ evolves according to the master equation

$$\frac{\mathrm{d}P(\boldsymbol{\sigma}; t)}{\mathrm{d}t} = \sum_\mu \sum_i \sum_{\langle ij \rangle} \Big[ w(\sigma_j^\mu, \sigma_i^\mu) P(\boldsymbol{\sigma}_{ij}^{\mu\mu}; t) - w(\sigma_i^\mu, \sigma_j^\mu) P(\boldsymbol{\sigma}; t) \Big], \tag{37}$$

where $\boldsymbol{\sigma}_{ij}^{\mu\mu}$ denotes the state obtained from $\boldsymbol{\sigma}$ by interchanging the spins $\sigma_i^\mu$ and $\sigma_j^\mu$ on lattice $\mu$. Note that these states only differ when $\sigma_i^\mu = -\sigma_j^\mu$. Given a fixed initial number of up spins in each lattice, denoted by $\{N_+^a, N_+^b\}$, the dynamics is confined to a configuration space of size $\binom{N}{N_+^a}\binom{N}{N_+^b}$. As in the case of single spin-flip dynamics, we choose the transition rates to satisfy detailed balance when $K_a = K_b$, yielding [51]

$$w(\sigma_i^\mu, \sigma_j^\mu) = \frac{1}{2\tau} \left[ 1 - \tanh\left( \frac{\Delta E_{ij}^{\mu\mu}}{2} \right) \right], \tag{38}$$

where $\Delta E_{ij}^{\mu\mu}$ is the change in energy after exchanging neighboring spins $\sigma_i^\mu$ and $\sigma_j^\mu$ on lattice $\mu \in \{a, b\}$, and $\tau \geq 0$ is the characteristic time scale for an intralattice exchange of spins. To compute $\Delta E_{ij}^{\mu\mu}$, we model the exchange as a sequence of two independent spin flips, where each spin flips in its own local field [67]. This approach ensures that the underlying free energy remains consistent with that of single spin-flip dynamics[3]. The local energy $E_i^\mu$ of spin $\sigma_i^\mu$ is given by Eq. (1), and the corresponding energy change is

$$\Delta E_{ij}^{\mu\mu} = (\sigma_i^\mu - \sigma_j^\mu)(h_i^\mu - h_j^\mu),$$

where $h_i^\mu$ is defined in Eq. (2). Note that upon interchanging two spins the new local fields of both spins also interchange, i.e., $h_i^\mu \leftrightarrow h_j^\mu$, and therefore $\Delta E_{ji}^{\mu\mu} = -\Delta E_{ij}^{\mu\mu}$. Using Eqs. (37) and (38), we derive the exact dynamical equation for the expectation value of a single spin defined in Eq. (6) [61, 67]:

$$\tau \frac{\mathrm{d}m_i^\mu(t)}{\mathrm{d}t} = \frac{1}{2} \sum_{\langle ij \rangle} \Big[ \langle (1 - \sigma_i^\mu \sigma_j^\mu) \tanh(h_i^\mu - h_j^\mu) \rangle(t) + m_j^\mu(t) - m_i^\mu(t) \Big]. \tag{39}$$

---

[3]Alternatively, one could take into account that the neighboring spin involved in the exchange does not contribute to the energy change during the flip. This leads to a slightly different expression for the underlying free energy w.r.t. single spin-flip dynamics as shown in [61]. However, this does not result in any qualitative changes.

As in Eq. (7), this equation is not closed due to the expectation value of $\tanh(h_i^\mu - h_j^\mu)$. To resolve this, we apply the MF approximation. Note that in [68] a similar MF approximation has been applied to study spin-exchange dynamics on a single lattice with exponential (Arrhenius-type) transition rates.

## 4.1 Mean-Field Approximation

We now evaluate Eq. (39) at the MF level, for which we use the approximation given by Eq. (8), along with a factorization of the pair correlations:

$$\langle \sigma_i^\mu \sigma_j^\mu \rangle \stackrel{\mathrm{MF}}{=} \langle \sigma_i^\mu \rangle \langle \sigma_j^\mu \rangle. \tag{40}$$

Inserting these approximations into Eq. (39) we obtain

$$\tau \frac{\mathrm{d}m_i^\mu(t)}{\mathrm{d}t} = \frac{1}{2} \sum_{\langle ij \rangle} \Big[ (1 - m_i^\mu(t)m_j^\mu(t)) \tanh\big(\langle h_i^\mu \rangle(t) - \langle h_j^\mu \rangle(t)\big) - m_i^\mu(t) + m_j^\mu(t) \Big]. \tag{41}$$

Using the identity given by Eq. (10) together with the following relation for $\Phi(x)$ given by Eq. (13)

$$2 \operatorname{arctanh}\left( \frac{m_i^\mu - m_j^\mu}{1 - m_i^\mu m_j^\mu} \right) = \frac{\mathrm{d}\Phi(m_i^\mu)}{\mathrm{d}m_i^\mu} - \frac{\mathrm{d}\Phi(m_j^\mu)}{\mathrm{d}m_j^\mu},$$

we can rewrite Eq. (41) as (omitting the $t$-dependence)

$$
\begin{aligned}
\tau \frac{\mathrm{d}m_i^a}{\mathrm{d}t} &= \frac{1}{2} \sum_{\langle ij \rangle} \mathcal{M}_{ij}^{aa}(\mathbf{m}^a, \mathbf{m}^b) \tanh\left( \frac{\partial \mathcal{F}(\mathbf{m}^a, \mathbf{m}^b)}{\partial m_j^a} - \frac{\partial \mathcal{F}(\mathbf{m}^a, \mathbf{m}^b)}{\partial m_i^a} + \frac{K_a - K_b}{2}(m_i^b - m_j^b) \right), \\
\tau \frac{\mathrm{d}m_i^b}{\mathrm{d}t} &= \frac{1}{2} \sum_{\langle ij \rangle} \mathcal{M}_{ij}^{bb}(\mathbf{m}^a, \mathbf{m}^b) \tanh\left( \frac{\partial \mathcal{F}(\mathbf{m}^a, \mathbf{m}^b)}{\partial m_j^b} - \frac{\partial \mathcal{F}(\mathbf{m}^a, \mathbf{m}^b)}{\partial m_i^b} - \frac{K_a - K_b}{2}(m_i^a - m_j^a) \right),
\end{aligned}
\tag{42}
$$

where the MF free energy $\mathcal{F}(\mathbf{m}^a, \mathbf{m}^b)$ is defined in Eq. (12), and the mobility $\mathcal{M}_{ij}^{\mu\mu}(\mathbf{m}^a, \mathbf{m}^b) \geq 0$ is given by

$$\mathcal{M}_{ij}^{\mu\mu}(\mathbf{m}^a, \mathbf{m}^b) \equiv 1 - m_i^\mu m_j^\mu - (m_i^\mu - m_j^\mu) \tanh\big(\langle h_i^\mu \rangle - \langle h_j^\mu \rangle\big).$$

Note that the external magnetic field $H_\mu$ cancels out in Eqs. (42), which is expected since the total magnetization in both lattices is conserved. For $N$ spins Eqs. (42) comprise a set of $2N$ nonlinearly coupled ordinary differential equations which can be solved numerically given suitable initial conditions.

## 4.2 Lyapunov Function for Reciprocal Interactions

When $K_a = K_b$ (i.e. for reciprocal interactions) the MF free energy is a Lyapunov function of Eqs. (42). To see this, we compute the time-derivative of $\mathcal{F}(\mathbf{m}^a, \mathbf{m}^b)$, which is given by

$$
\begin{aligned}
\tau \frac{\mathrm{d}\mathcal{F}}{\mathrm{d}t} &= \tau \sum_\mu \sum_{i=1}^N \frac{\partial \mathcal{F}}{\partial m_i^\mu} \frac{\mathrm{d}m_i^\mu}{\mathrm{d}t} \\
&\stackrel{(42)}{=} -\frac{1}{2} \sum_\mu \sum_{i=1}^N \sum_{\langle ij \rangle} \mathcal{M}_{ij}^{\mu\mu} \frac{\partial \mathcal{F}}{\partial m_i^\mu} \tanh\left( \frac{\partial \mathcal{F}}{\partial m_i^\mu} - \frac{\partial \mathcal{F}}{\partial m_j^\mu} \right) \\
&= -\frac{1}{4} \sum_\mu \sum_{i=1}^N \sum_{\langle ij \rangle} \mathcal{M}_{ij}^{\mu\mu} \left( \frac{\partial \mathcal{F}}{\partial m_i^\mu} - \frac{\partial \mathcal{F}}{\partial m_j^\mu} \right) \tanh\left( \frac{\partial \mathcal{F}}{\partial m_i^\mu} - \frac{\partial \mathcal{F}}{\partial m_j^\mu} \right) \leq 0,
\end{aligned}
\tag{43}
$$

456  where in the last line we used the symmetry $\mathcal{M}_{ij}^{\mu\mu} = \mathcal{M}_{ji}^{\mu\mu}$ and the last inequality follows from
457  the nonnegative mobilities and the fact that $x \tanh(x) \geq 0$ for $x \in \mathbb{R}$.

## 4.3  Thermodynamic Limit

459  Next, we determine the thermodynamic limit of Eqs. (42) (see Sect. 3.3). Taking $\{N_x, N_y\} \to \infty$
460  while keeping the system size $\{L_x, L_y\} = $ const., the lattice spacing vanishes, $\ell \to 0$. By
461  expanding in powers of $\ell$ up to second order and employing the gradient expansion (17),
462  Eqs. (42) yield the following partial differential equations (omitting the arguments $(\mathbf{x}, t)$):

$$
\begin{aligned}
\tau \frac{\partial m^a}{\partial t} &= \nabla \cdot \left( \frac{1 - (m^a)^2}{2} \nabla \left[ \frac{\delta \mathcal{F}[m^a, m^b]}{\delta m^a} - \frac{K_a - K_b}{2} m^b \right] \right), \\
\tau \frac{\partial m^b}{\partial t} &= \nabla \cdot \left( \frac{1 - (m^b)^2}{2} \nabla \left[ \frac{\delta \mathcal{F}[m^a, m^b]}{\delta m^b} + \frac{K_a - K_b}{2} m^a \right] \right),
\end{aligned}
\tag{44}
$$

463  where the MF free-energy functional $\mathcal{F}[m^a, m^b]$ is given by Eq. (19). Equations (44) are
464  immediately recognizable as nonreciprocally coupled Cahn-Hilliard equations [2, 19–21, 33–
465  35], with quadratic mobilities $[1 - (m^\mu)^2]/2$, which agrees with the mobilities in Eq. (22) up
466  to a constant factor of 1/2 which will be further explained in the next section. This derivation
467  thus provides a microscopic foundation for the nonreciprocal Cahn-Hilliard model. It is worth
468  noting that, unlike in Eq. (42), the hyperbolic tangent function $\tanh(\cdot)$ no longer explicitly
469  appears in Eq. (44). This absence results from a Taylor expansion of the hyperbolic tangent in
470  Eq. (42) to the smallest order in $\ell$, which is warranted by the outer sum, and yields a linear
471  approximation of the arguments. In contrast, for single spin-flip dynamics (see Eq. (18)),
472  such an expansion is not required because there is no outer sum, and the hyperbolic tangent
473  therefore remains present.

## 4.4  Linear Stability Analysis

475  To gain insight into the behavior of Eqs. (44), we perform a linear stability analysis of the
476  uniform steady state $m_0^\mu \in [-1, 1]$, which can be chosen freely and sets the total magnetization
477  in each lattice. We consider small perturbations of the form given by Eq. (24), where we
478  use $\tilde{\lambda}$ to distinguish the growth rate of exchange dynamics from those of the single spin-flip
479  dynamics, $\lambda$. Expanding Eqs. (44) to linear order in $\delta m^\mu$ yields the eigenvalue problem

$$
\tilde{\lambda} \begin{pmatrix} \delta m^a \\ \delta m^b \end{pmatrix} = \frac{|\mathbf{k}|^2}{2} \underline{\mathbf{L}} \begin{pmatrix} \delta m^a \\ \delta m^b \end{pmatrix},
$$

480  where the matrix $\underline{\mathbf{L}}$ is again given by Eq. (25). Hence, the growth rate $\tilde{\lambda}$ is related to the
481  eigenvalues of $\underline{\mathbf{L}}$ (given by Eq. (27)) via

$$
\tilde{\lambda} = \frac{|\mathbf{k}|^2}{2} \lambda,
\tag{45}
$$

482  and are shown in Fig. 4 for parameter values similar to those used in Fig. 2. The multiplicative
483  factor of $|\mathbf{k}|^2$ in Eq. (45) arises from the presence of the two conservation laws, which forces the
484  existence of a neutral mode ($\tilde{\lambda} = 0$) at $|\mathbf{k}| = 0$. The prefactor 1/2 reflects that spin exchange
485  arises from two independent single spin-flip events, effectively doubling the timescale com-
486  pared to single spin-flip dynamics. Since $\lambda$ and $\tilde{\lambda}$ are directly related via Eq. (45), $\mathrm{Re}(\lambda) = 0$

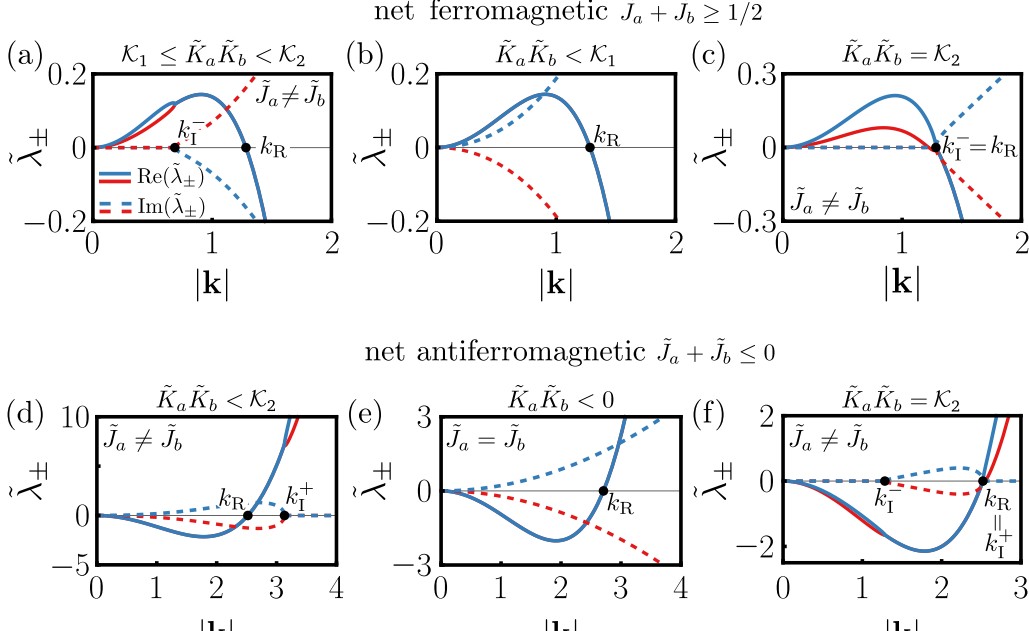

Figure 4: (a)-(f) Dispersion relations for intralattice spin-exchange dynamics [see Eq. (45)]. Panels (a)-(c) correspond to the net ferromagnetic regime with $\tilde{J}_a + \tilde{J}_b \geq 1/2$, and panels (d)-(f) correspond to the net antiferromagnetic regime with $\tilde{J}_a + \tilde{J}_b \leq 0$. In panels (a) and (d), a band of unstable stationary wavenumbers is observed, characterized by $\mathrm{Re}(\tilde{\lambda}_\pm) \geq 0$ and $\mathrm{Im}(\tilde{\lambda}_\pm) = 0$, along with an intermediate band of unstable oscillatory wavenumbers where $\mathrm{Re}(\tilde{\lambda}_\pm) \geq 0$ and $\mathrm{Im}(\tilde{\lambda}_\pm) > 0$. In panels (b) and (e), only unstable oscillatory wavenumbers are present. Panels (c) and (f) feature a critical exceptional point at $|\mathbf{k}| = k_R = k_I^\pm$, where $\mathrm{Re}(\tilde{\lambda}_\pm) = \mathrm{Im}(\tilde{\lambda}_\pm) = 0$. The auxiliary functions $\mathcal{K}_1(\tilde{J}_a, \tilde{J}_b)$ and $\mathcal{K}_2(\tilde{J}_a, \tilde{J}_b)$ are defined in Eqs. (28)-(29), and the wavenumbers $k_I^\pm$ and $k_R$ are given by Eqs. (31)-(32). Parameter values used in each panel are chosen as in Fig. 2.

and/or $\mathrm{Im}(\lambda) = 0$ directly imply $\mathrm{Re}(\tilde{\lambda}) = 0$ and/or $\mathrm{Im}(\tilde{\lambda}) = 0$. Consequently, the onset of the Hopf and Allen-Cahn instabilities identified in Sect. 3.5 for $\lambda$ also apply to $\tilde{\lambda}$ [33]. In particular, the bands of unstable wavenumbers for oscillatory and stationary modes are identical to the ones in Sect. 3.5 as even the wavenumber values $k_I^\pm$ and $k_R$ given by Eqs. (31)-(32) remain the same. In the presence of two conservation laws, the Hopf instability becomes a conserved Hopf instability, while the Allen-Cahn instability becomes a Cahn-Hilliard instability.

Upon incorporating higher-order gradient terms into the free-energy functional (19) (see Sect. 7) to regularize the antiferromagnetic divergence of the eigenvalues in the limit $|\mathbf{k}| \to \infty$ (see Fig. 4(d)-(f)), one obtains two nonreciprocally coupled phase-field crystal (PFC) equations [69] (also called nonreciprocally coupled conserved Swift-Hohenberg equations). The dispersion relations with the ultraviolet catastrophe would then become physically well-defined cases of conserved-Turing or conserved-wave instabilities [35].

### 4.5 Spurious Gradient Dynamics

A notable feature of Eqs. (44) is that they can be recast into the form of a *spurious* gradient dynamics [37] (omitting the arguments $(\mathbf{x}, t)$):

$$
\begin{aligned}
\tau \frac{\partial m^a}{\partial t} &= \nabla \cdot \left( \frac{1 - (m^a)^2}{2(1 + K_b/K_a)} \nabla \left[ \frac{\delta \hat{\mathcal{F}}[m^a, m^b]}{\delta m^a} \right] \right), \\
\tau \frac{\partial m^b}{\partial t} &= \nabla \cdot \left( \frac{1 - (m^b)^2}{2(1 + K_a/K_b)} \nabla \left[ \frac{\delta \hat{\mathcal{F}}[m^a, m^b]}{\delta m^b} \right] \right),
\end{aligned}
\tag{46}
$$

where the *spurious* free-energy functional is given by

$$
\hat{\mathcal{F}}[m^a, m^b] = \frac{1}{2} \int d\mathbf{x} \left[ \hat{f}(m^a, m^b) + \sum_\mu \left( 1 + K_\nu/K_\mu \right) J_\mu |\nabla m^\mu|^2 \right], \quad \nu \neq \mu,
$$

and the local *spurious* free-energy density reads

$$
\hat{f}(m^a, m^b) \equiv \sum_\mu \left( 1 + K_\nu/K_\mu \right) \left[ \Phi(m^\mu) - 2H_\mu m^\mu - 4J_\mu (m^\mu)^2 \right] - (K_a + K_b) m^a m^b, \quad \nu \neq \mu.
$$

The dynamics in Eqs. (46) are referred to as *spurious* gradient dynamics because the effective mobilities become negative when $K_a/K_b < -1$, which is in violation of basic thermodynamic principles [37]. It is important to note that the apparent divergence of the mobilities at $K_a = -K_b$ is canceled by the vanishing prefactor in front of the local spurious free-energy density, as can be seen by applying L'Hôpital's rule[4]. Therefore, the dynamics remain well-defined at this parameter value. In future work, we will exploit the spurious gradient form to systematically chart the phase diagram of the nonreciprocal Ising model with two conserved order parameters, cf. [37].

## 5 One Conservation Law: Inter- and Intralattice Exchange Dynamics

Finally, we consider the scenario in which nearest-neighbor spins can interchange both within and between the two lattices, as illustrated in Figs. 1(c) and 1(d), respectively. Under this dynamics, the sum of the total magnetizations, $M^a(t) + M^b(t)$, is conserved. Consequently, we anticipate that the system is governed by a nonreciprocal reactive CH model with one conservation law. The probability $P(\boldsymbol{\sigma}; t)$ for the system to be in spin configuration $\boldsymbol{\sigma}$ at time $t$ evolves according to the master equation

$$
\begin{aligned}
\frac{dP(\boldsymbol{\sigma}; t)}{dt} &= \sum_i \left[ w(\sigma_i^b, \sigma_i^a) P(\boldsymbol{\sigma}_{ii}^{ab}; t) - w(\sigma_i^a, \sigma_i^b) P(\boldsymbol{\sigma}; t) \right] \\
&\quad + \sum_\mu \sum_i \sum_{\langle ij \rangle} \left[ w(\sigma_j^\mu, \sigma_i^\mu) P(\boldsymbol{\sigma}_{ij}^{\mu\mu}; t) - w(\sigma_i^\mu, \sigma_j^\mu) P(\boldsymbol{\sigma}; t) \right],
\end{aligned}
\tag{47}
$$

---

[4]Let $x \equiv K_a/K_b$. Then,

$$
\lim_{x \to -1} \frac{1 + x}{1 + x} = 1, \quad \text{and} \quad \lim_{x \to -1} \frac{1 + 1/x}{1 + x} = -1.
$$

Hence, no diverging terms remain at $K_a = -K_b$ in Eqs. (46).

where $\boldsymbol{\sigma}_{ii}^{ab}$ denotes the spin configuration obtained from $\boldsymbol{\sigma}$ by exchanging spins $\sigma_i^a$ and $\sigma_i^b$. The first term on the right-hand side (r.h.s.) of Eq. (47) accounts for interlattice spin exchange, while the second term corresponds to intralattice exchange, as previously introduced in Eq. (37). For a fixed initial number of up spins in each lattice, denoted by $\{N_+^a, N_+^b\}$, the dynamics is confined to a configuration space of size $\binom{2N}{N_+^a + N_+^b}$. The transition rate for intralattice exchange is given by Eq. (38), whereas the interlattice exchange is governed by

$$w(\sigma_i^a, \sigma_i^b) = \frac{1}{2\tau}\left[1 - \tanh\left(\frac{\Delta E_{ii}^{ab}}{2}\right)\right], \tag{48}$$

where $\Delta E_{ii}^{ab}$ denotes the change in local energy resulting from the exchange of $\sigma_i^a$ and $\sigma_i^b$, and $\tau > 0$ is an intrinsic timescale for the interlattice exchange of spins. For convenience we keep the intrinsic timescales for inter- and intralattice exchange equal. As in Sect. 4, we assume that interlattice spin exchange proceeds via two independent single spin-flip events. Under this assumption, the energy change becomes

$$\Delta E_{ii}^{ab} = (\sigma_i^a - \sigma_i^b)(h_i^a - h_i^b),$$

which vanishes when $\sigma_i^a = \sigma_i^b$, since an exchange of equivalent spins does not change the state. In the case of perfect nonreciprocity with $K_a = -K_b = K$ we find that the nonreciprocal coupling vanishes in (48), since:

$$h_i^a - h_i^b = H_a - H_b + J_a \sum_{\langle ij \rangle} \sigma_j^a - J_b \sum_{\langle ij \rangle} \sigma_j^b + K(\sigma_i^b + \sigma_i^a),$$

where the last term vanishes because the transition only takes place when $\sigma_i^a = -\sigma_i^b$. Combining Eqs. (47) and (48), one can derive the following exact expression for the time evolution of the single-spin expectation value (omitting the $t$-dependence in the notation):

$$\tau \frac{dm_i^\mu}{dt} = \frac{1}{2}\left(\langle(1 - \sigma_i^\mu \sigma_i^\nu)\tanh(h_i^\mu - h_i^\nu)\rangle - m_i^\mu + m_i^\nu\right)$$
$$+ \frac{1}{2}\sum_{\langle ij \rangle}\left[\langle(1 - \sigma_i^\mu \sigma_j^\mu)\tanh(h_i^\mu - h_j^\mu)\rangle - m_i^\mu + m_j^\mu\right], \quad \nu \neq \mu. \tag{49}$$

Our next task is to close Eq. (49) using the MF approximation.

## 5.1 Mean-Field Approximation

We now evaluate Eq. (49) at the MF level, using the approximations given by Eqs. (8) and (40), followed by the application of the exact identity in Eq. (10). This yields the following expression (omitting the $t$-dependence in the notation):

$$\tau \frac{dm_i^a}{dt} = \frac{1}{2}\sum_{\langle ij \rangle}\mathcal{M}_{ij}^{aa}\tanh\left(\frac{\partial \mathcal{F}(\mathbf{m}^a, \mathbf{m}^b)}{\partial m_j^a} - \frac{\partial \mathcal{F}(\mathbf{m}^a, \mathbf{m}^b)}{\partial m_i^a} + \frac{K_a - K_b}{2}(m_i^b - m_j^b)\right) + \mathcal{R}_i(\mathbf{m}^a, \mathbf{m}^b),$$
$$\tau \frac{dm_i^b}{dt} = \frac{1}{2}\sum_{\langle ij \rangle}\mathcal{M}_{ij}^{bb}\tanh\left(\frac{\partial \mathcal{F}(\mathbf{m}^a, \mathbf{m}^b)}{\partial m_j^b} - \frac{\partial \mathcal{F}(\mathbf{m}^a, \mathbf{m}^b)}{\partial m_i^b} - \frac{K_a - K_b}{2}(m_i^a - m_j^a)\right) - \mathcal{R}_i(\mathbf{m}^a, \mathbf{m}^b),$$
$$\tag{50}$$

where the first term on the r.h.s. originates from intralattice spin exchange and corresponds to the r.h.s. of Eqs. (42). The free energy $\mathcal{F}(\mathbf{m}^a, \mathbf{m}^b)$ with $\mathbf{m}^\mu = \{m_1^\mu, ..., m_N^\mu\}$ is given by Eq. (12),

and the second term, $\mathcal{R}_i(\mathbf{m}^a, \mathbf{m}^b)$, arises from interlattice exchange and reads

$$\mathcal{R}_i(\mathbf{m}^a, \mathbf{m}^b) \equiv \frac{\mathcal{M}_{ii}^{ab}(\mathbf{m}^a, \mathbf{m}^b)}{2} \tanh\left( \frac{\partial \mathcal{F}(\mathbf{m}^a, \mathbf{m}^b)}{\partial m_i^b} - \frac{\partial \mathcal{F}(\mathbf{m}^a, \mathbf{m}^b)}{\partial m_i^a} + \frac{K_a - K_b}{2}(m_i^a + m_i^b) \right), \quad (51)$$

with the interlattice exchange mobility given by

$$\mathcal{M}_{ii}^{ab}(\mathbf{m}^a, \mathbf{m}^b) \equiv 1 - m_i^a m_i^b - (m_i^a - m_i^b) \tanh(\langle h_i^a \rangle - \langle h_i^b \rangle), \quad (52)$$

which is nonnegative. An interesting consequence of Eq. (51) is the breakdown of the afore-mentioned property related to the nonreciprocal coupling: for perfectly nonreciprocal couplings, Eq. (51) is not independent of the nonreciprocal coupling strength, whereas Eq. (49) is. This discrepancy arises due to the MF approximation, which replaces discrete spin variables $\sigma_i^\mu$ with continuous fields $m_i^\mu$. As a result, the approximation neglects the fact that certain discrete-spin interactions may identically vanish in specific configurations. Finally, note that the external magnetic field $H_\mu$, as expected, only enters in the reaction term $\mathcal{R}_i(\mathbf{m}^a, \mathbf{m}^b)$. For a system of $N$ spins, Eqs. (50) form a set of $2N$ nonlinearly coupled ordinary differential equations, which can be solved numerically given appropriate initial conditions.

## 5.2 Lyapunov Function for Reciprocal Interactions

When $K_a = K_b$ (i.e., in the case of reciprocal interactions), the MF free energy $\mathcal{F}(\mathbf{m}^a, \mathbf{m}^b)$ serves as a Lyapunov function for the dynamics governed by Eqs. (50). To demonstrate this, we evaluate the time derivative of $\mathcal{F}(\mathbf{m}^a, \mathbf{m}^b)$. For the intralattice exchange contribution, we can directly apply the bound given by Eq. (43), yielding the result

$$\tau \frac{d\mathcal{F}}{dt} = \tau \sum_\mu \sum_{i=1}^N \frac{\partial \mathcal{F}}{\partial m_i^\mu} \frac{dm_i^\mu}{dt} \overset{(50)}{\underset{(43)}{\leq}} -\frac{1}{2} \sum_{i=1}^N \mathcal{M}_{ii}^{ab} \left( \frac{\partial \mathcal{F}}{\partial m_i^a} - \frac{\partial \mathcal{F}}{\partial m_i^b} \right) \tanh\left( \frac{\partial \mathcal{F}}{\partial m_i^a} - \frac{\partial \mathcal{F}}{\partial m_i^b} \right) \leq 0,$$

where the final inequality follows from the identity $x \tanh(x) \geq 0$ for all $x \in \mathbb{R}$ and the non-negative mobilities.

## 5.3 Thermodynamic Limit

Next, we determine the thermodynamic limit of Eqs. (50) (see Sect. 3.3). Taking $\{N_x, N_y\} \to \infty$ while keeping the system size $\{L_x, L_y\} = \text{const.}$, the lattice spacing vanishes, $\ell \to 0$. By expanding in powers of $\ell$ up to second order and employing the gradient expansion (17), Eqs. (50) yield the following partial differential equations (omitting $(\mathbf{x}, t)$ arguments):

$$\tau \frac{\partial m^a}{\partial t} = \nabla \cdot \left( \frac{1 - (m^a)^2}{2} \nabla \left[ \frac{\delta \mathcal{F}[m^a, m^b]}{\delta m^a} - \frac{K_a - K_b}{2} m^b \right] \right) + \mathcal{R}(m^a, m^b),$$

$$\tau \frac{\partial m^b}{\partial t} = \nabla \cdot \left( \frac{1 - (m^b)^2}{2} \nabla \left[ \frac{\delta \mathcal{F}[m^a, m^b]}{\delta m^b} + \frac{K_a - K_b}{2} m^a \right] \right) - \mathcal{R}(m^a, m^b),$$

$$\quad (53)$$

where $\mathcal{F}[m^a, m^b]$ is given by Eq. (19), the reaction term $\mathcal{R}(m^a, m^b)$ is the thermodynamic limit of Eq. (51)

$$\mathcal{R}(m^a, m^b) \equiv \frac{\mathcal{M}^{ab}(m^a, m^b)}{2} \tanh\left( \frac{\delta \mathcal{F}[m^a, m^b]}{\delta m^b} - \frac{\delta \mathcal{F}[m^a, m^b]}{\delta m^a} + \frac{K_a - K_b}{2}(m^a + m^b) \right), \quad (54)$$

and the mobility term $\mathcal{M}^{ab}(m^a, m^b)$ is the thermodynamic limit of Eq. (52)

$$\mathcal{M}^{ab}(m^a, m^b) \equiv 1 - m^a m^b - (m^a - m^b)\tanh\left(\sum_\mu (-1)^{\delta_{\mu,b}}\left[H_\mu + J_\mu[4 + \nabla^2]m^\mu + K_\mu m^\nu\right]\right), \tag{55}$$

with $\delta_{\mu,b} = 1$ when $\mu = b$ and 0 otherwise. From the structure of Eqs. (53), we identify two nonreciprocally coupled Cahn-Hilliard equations with an additional nonreciprocal reactive coupling that conserves the sum of both total magnetizations, similar to other Cahn-Hilliard models with overall mass-conserving reaction terms [30–32] that themselves extend reaction-diffusion models with mass conservation [38, 40, 45, 70] towards nonideal systems [71]. A notable feature of the reaction term (54) is its strong dependence on the underlying free energy functional $\mathcal{F}[m^a, m^b]$, analogous to the role of the underlying free energy in the kinetics of mass action reactions for nonideal systems [71] and reactive thin-film hydrodynamics [72, 73].

## 5.4 Expansion Close To Stationary States

Similar to our analysis in Sec. 3.4, we note that near stationary states the argument of the hyperbolic tangent in Eq. (54) is small. Let us recall $x^\mu$ is given by

$$x^\mu(m^a, m^b) \equiv \frac{\delta \mathcal{F}[m^a, m^b]}{\delta m^\mu} + (-1)^{\delta_{\mu,a}}\frac{K_a - K_b}{2}m^\nu, \quad \nu \neq \mu, \tag{56}$$

so that $|x^\mu| \ll 1$ close to stationarity. The hyperbolic tangent in Eq. (54) can then be expanded as

$$\tanh(x^b - x^a) = x^b - x^a + \mathcal{O}((x^b - x^a)^3).$$

At the same time, we expand the mobility in Eq. (55) in powers of $x^\mu$. Using Eq. (10), Eq. (55) can be rewritten (and thus expanded) as

$$\mathcal{M}^{ab}(m^a, m^b) = 1 - m^a m^b - (m^a - m^b)\tanh\left(\text{arctanh}([m^a - m^b]/[1 - m^a m^b]) + x^b - x^a\right)$$

$$= \frac{[1 - (m^a)^2][1 - (m^b)^2]}{1 - m^a m^b} + \mathcal{O}(x^b - x^a).$$

Consequently, sufficiently close to stationarity, Eqs. (18) reduce (to linear order in $x^\mu$) to the following (much simpler) equations:

$$\tau\frac{\partial m^a}{\partial t} \simeq \nabla\cdot\left(\frac{1 - (m^a)^2}{2}\nabla x^a(m^a, m^b)\right) + \frac{[1 - (m^a)^2][1 - (m^b)^2]}{1 - m^a m^b}\left(x^b(m^a, m^b) - x^a(m^a, m^b)\right),$$

$$\tau\frac{\partial m^b}{\partial t} \simeq \nabla\cdot\left(\frac{1 - (m^b)^2}{2}\nabla x^b(m^a, m^b)\right) - \frac{[1 - (m^a)^2][1 - (m^b)^2]}{1 - m^a m^b}\left(x^b(m^a, m^b) - x^a(m^a, m^b)\right),$$

where $x^\mu(m^a, m^b)$ is given by Eq. (56).

## 5.5 Linear Stability Analysis

To understand the conditions under which small perturbations grow or decay, we analyze the linear stability of uniform solutions to Eqs. (53). We consider small perturbations of the form given in Eq. (24), where the uniform state $m_0^\mu$ must lie on the reactive nullcline [44], defined by

$$\mathcal{R}(m_0^a, m_0^b) = 0. \tag{57}$$

In Fig. 5(a)-(c) we show the reactive nullcline (see the red line) for various parameter values. One specific symmetric state on the reactive nullcline is given by

$$m_0^a = m_0^b = \frac{H_a - H_b}{4(J_b - J_a) + K_b - K_a}, \quad \text{for} \quad \left| \frac{H_a - H_b}{4(J_b - J_a) + K_b - K_a} \right| \leq 1, \tag{58}$$

which is shown as the red point in Fig. 5(a)-(c). Quite surprisingly, this symmetric solution even exists in the presence of two unequal magnetic field $H_a \neq H_b$.

We denote the growth rate of the perturbations by $\hat{\lambda}$ to distinguish them from those in Eqs. (27) and (45). Linearizing Eqs. (53) around the uniform steady state leads to the eigenvalue problem:

$$\hat{\lambda} \begin{pmatrix} \delta m^a \\ \delta m^b \end{pmatrix} = \left( \frac{|\mathbf{k}|^2}{2} \underline{\mathbf{L}} + \underline{\mathbf{R}} \right) \begin{pmatrix} \delta m^a \\ \delta m^b \end{pmatrix},$$

where $\underline{\mathbf{L}}$ is defined in Eq. (25), and $\underline{\mathbf{R}}$ is the Jacobian of the reaction term, evaluated at the uniform steady state:

$$\underline{\mathbf{R}} \equiv \frac{1/\tau}{1 - m_0^a m_0^b} \begin{pmatrix} [1 - (m_0^b)^2](\tilde{J}_a(4 - |\mathbf{k}|^2) - \tilde{K}_b - 1) & -[1 - (m_0^a)^2](\tilde{J}_b(4 - |\mathbf{k}|^2) - \tilde{K}_a - 1) \\ -[1 - (m_0^b)^2](\tilde{J}_a(4 - |\mathbf{k}|^2) - \tilde{K}_b - 1) & [1 - (m_0^a)^2](\tilde{J}_b(4 - |\mathbf{k}|^2) - \tilde{K}_a - 1) \end{pmatrix},$$

with $\tilde{J}_\mu$ and $\tilde{K}_\mu$ defined in Eq. (26). In contrast to the analysis in Sect. 4.4, the growth $\hat{\lambda}$ is not related in a simple way to the eigenvalues of $\underline{\mathbf{L}}$, as the reaction term introduces a nontrivial contribution. We therefore focus on the special case of the symmetric uniform steady state given by Eq. (58), leaving a more detailed and systematic analysis of the complete spectral properties for future work. In this case, the eigenvalues of $|\mathbf{k}|^2 \underline{\mathbf{L}}/2 + \underline{\mathbf{R}}$ (and therefore the growth rate) read

$$\hat{\lambda}_\pm = \frac{1}{2\tau} \left( \Theta \pm \sqrt{\Theta^2 - 4\Delta} \right), \tag{59}$$

where the trace $\Theta$ and determinant $\Delta$ are given by

$$\Theta \equiv (2 + |\mathbf{k}|^2)[(\tilde{J}_a + \tilde{J}_b)(4 - |\mathbf{k}|^2) - 2]/2 - \tilde{K}_a - \tilde{K}_b,$$
$$\Delta \equiv |\mathbf{k}|^2(4 + |\mathbf{k}|^2)\left([1 - \tilde{J}_a(4 - |\mathbf{k}|^2)][1 - \tilde{J}_b(4 - |\mathbf{k}|^2)] - \tilde{K}_a \tilde{K}_b\right).$$

Observe that $\Delta = 0$ at $|\mathbf{k}| = 0$, and hence one eigenvalue $\hat{\lambda}_\pm$ always vanishes at this point, as expected in the presence of one conservation law. The second eigenvalue can be positive or negative at $|\mathbf{k}| = 0$ depending on the sign of $\Theta$, with associated eigenvector $(-1, 1)^\mathrm{T}$ corresponding to uniform magnetization redistribution between the two lattices. The second eigenvalue also vanishes at $|\mathbf{k}| = 0$ when $\Theta = 0$, which occurs precisely for

$$\hat{\lambda}_\pm \big|_{k=0} = 0 \implies 4(\tilde{J}_a + \tilde{J}_b) = 2 + \tilde{K}_a + \tilde{K}_b. \tag{60}$$

The corresponding eigenvectors are degenerate, taking the form $(-1, 1)^\mathrm{T}$. Equation (60) thus defines the critical threshold between stable and unstable magnetization redistribution modes:

- For $4(\tilde{J}_a + \tilde{J}_b) > 2 + \tilde{K}_a + \tilde{K}_b$ we have $\lambda_+ > 0$ and $\lambda_- = 0$ at $|\mathbf{k}| = 0$, indicating that uniform magnetization redistribution amplifies over time. This is shown in Fig. 5(c), where the symmetric uniform state (red point) is unstable at $|\mathbf{k}| = 0$ and therefore evolves along the line $m_0^a + m_0^b = \text{const.}$ towards one of two stables uniform states indicated with the black points.

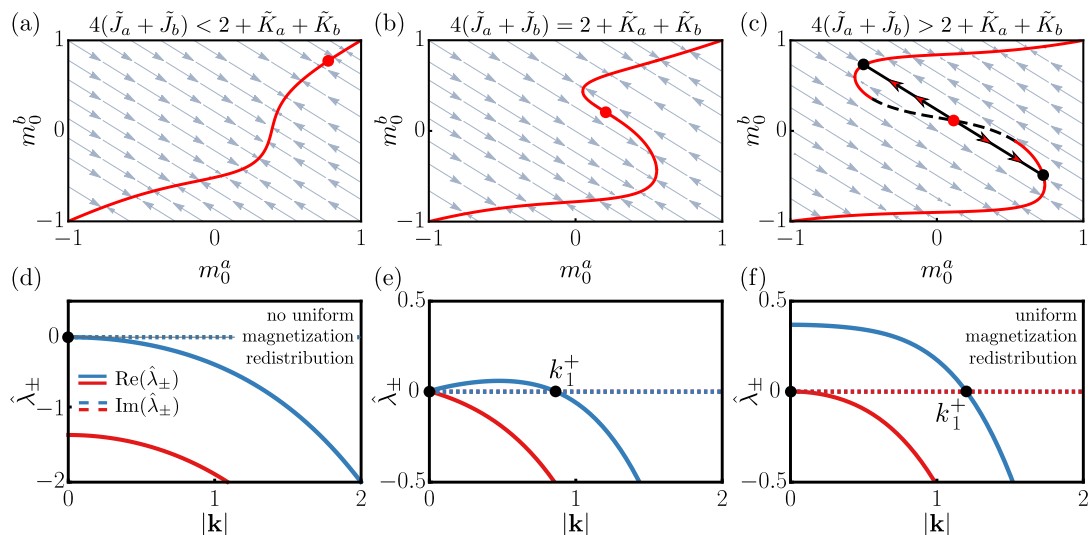

Figure 5: (a)–(c) Reactive nullcline (red line) defined by Eq. (57), shown for parameter values (a) below, (b) at, and (c) above the critical threshold given in Eq. (60). The background illustrates the flow field for the uniform state, as governed by Eqs. (53). In panel (c), the black dashed segment marks parts of the nullcline that are linearly unstable with respect to uniform perturbations. The red points indicate the symmetric uniform steady states [Eq. (58)], which is unstable in (c) and therefore evolves along the line $m_0^a + m_0^b = $ const. towards one of two stable states (black points). (d)–(f) Dispersion relations corresponding to linear perturbations around the symmetric uniform steady state, as given by Eq. (59). In panel (d), both eigenvalues are non-positive at $|\mathbf{k}| = 0$, indicating linear stability. In panel (e), both eigenvalues vanish at $|\mathbf{k}| = 0$, marking the onset of a uniform instability. In panel (f), one eigenvalue is positive at $|\mathbf{k}| = 0$, indicating a stationary instability of a band of harmonic modes. The zero crossing at $k_1^+$ is given by Eq. (34). Parameter values $(J_a, J_b, K_a, K_b, H_a, H_b)$ used in each panel are given by: (a, d) $(0.2, 0.25, 0.05, -0.05, 0.04, -0.04)$, (b, e) $(0.2, 0.323031, 0.05, -0.05, 0.04, -0.04)$, (c, f) $(0.2, 0.4, 0.05, -0.05, 0.04, -0.04)$. Note that at $(0.2, 0.279083, 0.05, -0.05, 0.04, -0.04)$ a Cahn-Hilliard instability occurs, which is not explicitly shown.

- In contrast, for $4(\tilde{J}_a + \tilde{J}_b) < 2 + \tilde{K}_a + \tilde{K}_b$ we have $\lambda_+ = 0$ and $\lambda_- < 0$ at $|\mathbf{k}| = 0$, implying that uniform magnetization redistribution decays. This is shown in Fig. 5(a), where the symmetric uniform state (red point) is stable. In Fig. 5(b) the symmetric uniform state is at the onset of becoming unstable, as the slope of the reactive nullcline at this point is tangent to the constant magnetization line $m_0^a + m_0^b = $ const.

Note that before the onset of the uniform instability there is the onset of a Cahn-Hilliard instability after which a range of wavenumbers $0 < k < k_1^+$ with $k_1^+$ given Eq. (34) become unstable. The onset of the Cahn-Hilliard instability occurs for

$$\left.\frac{\mathrm{d}\hat{\lambda}_+}{\mathrm{d}k^2}\right|_{k=0} = 0 \implies (4\tilde{J}_a - 1)(4\tilde{J}_b - 1) = \tilde{K}_a \tilde{K}_b.$$

Figure 5(d)-(f) illustrates the dispersion relations corresponding to values below, at, and above this critical threshold. Note that the parameter threshold defined by Eq. (60) admits a clear physical interpretation. Consider the uniform symmetric steady state where $m_0^a = m_0^b = 0$ for $H_a = H_b$, corresponding to an equal number of up and down spins in each lattice. For

Table 1: The sixteen distinct nonreciprocal partial differential equation models obtained when combining the dynamical processes in the two lattices in all possible ways. Only options are included where both lattices contribute to the dynamics. Note that additional dynamics obtained when interchanging the two lattices are not included. The terms $\mathcal{A}^\mu$, $\mathcal{B}^\mu$, and $\mathcal{R}$ are explained in the main text.

| # | single flip | | intra-ex. | | inter-ex. | dynamical equations | | conservation |
|---|---|---|---|---|---|---|---|---|
| | a | b | a | b | | $\tau\partial_t m^a =$ | $\tau\partial_t m^b =$ | laws |
| 1 | x | x | - | - | - | $\mathcal{A}^a$ | $\mathcal{A}^b$ | 0 |
| 2 | x | x | - | - | x | $\mathcal{A}^a + \mathcal{R}$ | $\mathcal{A}^b - \mathcal{R}$ | 0 |
| 3 | x | - | - | x | x | $\mathcal{A}^a + \mathcal{R}$ | $\mathcal{B}^b - \mathcal{R}$ | 0 |
| 4 | x | - | - | - | x | $\mathcal{A}^a + \mathcal{R}$ | $-\mathcal{R}$ | 0 |
| 5 | x | x | x | x | - | $\mathcal{A}^a + \mathcal{B}^a$ | $\mathcal{A}^b + \mathcal{B}^b$ | 0 |
| 6 | x | x | x | - | - | $\mathcal{A}^a + \mathcal{B}^a$ | $\mathcal{A}^b$ | 0 |
| 7 | x | x | x | x | x | $\mathcal{A}^a + \mathcal{B}^a + \mathcal{R}$ | $\mathcal{A}^b + \mathcal{B}^b - \mathcal{R}$ | 0 |
| 8 | x | x | x | - | x | $\mathcal{A}^a + \mathcal{B}^a + \mathcal{R}$ | $\mathcal{A}^b - \mathcal{R}$ | 0 |
| 9 | x | - | x | x | x | $\mathcal{A}^a + \mathcal{B}^a + \mathcal{R}$ | $\mathcal{B}^b - \mathcal{R}$ | 0 |
| 10 | x | - | x | - | x | $\mathcal{A}^a + \mathcal{B}^a + \mathcal{R}$ | $-\mathcal{R}$ | 0 |
| 11 | - | - | x | x | x | $\mathcal{B}^a + \mathcal{R}$ | $\mathcal{B}^b - \mathcal{R}$ | 1 |
| 12 | - | - | x | - | x | $\mathcal{B}^a + \mathcal{R}$ | $-\mathcal{R}$ | 1 |
| 13 | x | - | - | x | - | $\mathcal{A}^a$ | $\mathcal{B}^b$ | 1 |
| 14 | x | - | x | x | - | $\mathcal{A}^a + \mathcal{B}^a$ | $\mathcal{B}^b$ | 1 |
| 15 | - | - | - | - | x | $\mathcal{R}$ | $-\mathcal{R}$ | 1 |
| 16 | - | - | x | x | - | $\mathcal{B}^a$ | $\mathcal{B}^b$ | 2 |

positive coupling strengths $J_a$ and $J_b$, it becomes energetically favorable to segregate the spin states, placing all up spins in one lattice and all down spins in the other. When the values of $J_a$ and $J_b$ exceed the critical threshold in Eq. (60), this energetic preference outweighs the entropic cost of demixing. As a result, the system undergoes a spontaneous redistribution of magnetization between the lattices, and each spin species preferentially occupies a distinct lattice. This mechanism underlies the instability of uniform magnetization redistribution and drives the emergence of demixed configurations between the two lattices.

# 6 Further Combinations of Spin-Flip and Spin-Exchange Dynamics

In the previous section, we have shown that the combination of inter- and intralattice exchange dynamics leads to additive contributions in the resulting partial differential equations. This additivity holds for all three types of kinetic updates of the considered Ising lattices, namely, single spin-flip dynamics and both intra- and interlattice spin-exchange dynamics. Consequently, partial differential equations corresponding to any combination of allowed kinetic updates among the two lattices can be constructed by appropriately combining the following three fundamental terms:

- Single spin-flip dynamics:

$$\tau \frac{\partial m^\mu}{\partial t} = \mathcal{A}^\mu \equiv -\mathcal{M}^\mu(m^a, m^b)\tanh\left(\frac{\delta\mathcal{F}[m^a, m^b]}{\delta m^\mu} - (-1)^{\delta_{\mu,b}}\frac{K_a - K_b}{2}m^\nu\right),$$

with $\nu \neq \mu$, the free energy $\mathcal{F}[m^a, m^b]$ given by Eq. (19), and mobility $\mathcal{M}^\mu(m^a, m^b)$ by Eq. (21). We abbreviate the r.h.s. as $\mathcal{A}^\mu$ as the linearization of its reciprocal limit corresponds to a noiseless model-A in the Hohenberg-Halperin classification [74].

- Intralattice spin-exchange dynamics:

$$\tau \frac{\partial m^\mu}{\partial t} = \mathcal{B}^\mu \equiv \nabla \cdot \left( \frac{1 - (m^\mu)^2}{2} \nabla \left[ \frac{\delta \mathcal{F}[m^a, m^b]}{\delta m^\mu} - (-1)^{\delta_{\mu,b}} \frac{K_a - K_b}{2} m^\nu \right] \right),$$

again with $\nu \neq \mu$, and the free energy given by Eq. (19). We abbreviate the r.h.s. as $\mathcal{B}^\mu$ as its reciprocal limit corresponds to a noiseless model-B in the classification of [74].

- Interlattice spin-exchange dynamics:

$$\tau \frac{\partial m^\mu}{\partial t} = (-1)^{\delta_{\mu,b}} \mathcal{R}$$
$$= (-1)^{\delta_{\mu,b}} \frac{\mathcal{M}^{ab}(m^a, m^b)}{2} \tanh\left( \frac{\delta \mathcal{F}[m^a, m^b]}{\delta m^b} - \frac{\delta \mathcal{F}[m^a, m^b]}{\delta m^a} + \frac{K_a - K_b}{2}(m^a + m^b) \right),$$

where the mobility $\mathcal{M}^{ab}(m^a, m^b)$ is given by Eq. (55).

As an illustrative example, consider a scenario where single spin-flip dynamics is applied on lattice $a$, while intralattice spin-exchange governs the dynamics on lattice $b$. For such dynamics, the total magnetization in lattice $b$ is conserved, resulting in one conservation law (see also row 13 in Table 1). Combining both rules, we obtain a nonreciprocally coupled Allen-Cahn and Cahn-Hilliard system which takes the form (omitting the arguments $(\mathbf{x}, t)$)

$$\tau \frac{\partial m^a}{\partial t} = \mathcal{A}^a = -\mathcal{M}^a(m^a, m^b) \tanh\left( \frac{\delta \mathcal{F}[m^a, m^b]}{\delta m^a} - \frac{K_a - K_b}{2} m^b \right),$$
$$\tau \frac{\partial m^b}{\partial t} = \mathcal{B}^b = \nabla \cdot \left( \frac{1 - (m^b)^2}{2} \nabla \left[ \frac{\delta \mathcal{F}[m^a, m^b]}{\delta m^b} + \frac{K_a - K_b}{2} m^a \right] \right).$$

The reciprocal limit of this coupled model corresponds to a noiseless model-C in the classification of [74, 75] as introduced in [76] to investigate systems with first-order phase transitions in the presence of a single conservation law. These models have also found application in understanding phase separation and gelation phenomena in cellular fluids [77, 78], and various studies have focused on developing numerical methods to solve such systems [79, 80].

Going beyond this specific example, one can construct sixteen distinct nonreciprocal partial differential equation models that incorporate any combination of dynamical processes in the two lattices. A corresponding list is given in Table 1. Of these sixteen models for two (nonreciprocally) coupled fields, ten describe cases with no conservation laws, five possess one conservation law, and one — the nonreciprocal Cahn–Hilliard model — has two conservation laws.

## 7 Conclusion

In this work, we have derived the macroscopic continuum field equations for two nonreciprocally coupled scalar fields with zero, one, and two conservation laws, directly from a microscopic model consisting of two nonreciprocally coupled Ising lattices with different types

of single spin-flip and spin-exchange dynamics within and between the lattices. By employing the mean-field approximation and taking the thermodynamic limit, we obtained three distinct classes of dynamical equations: the nonreciprocal Allen-Cahn model (18) (zero conservation laws) for single spin-flip dynamics, the nonreciprocal Cahn-Hilliard model (44) (two conservation laws) for intralattice spin-exchange dynamics, and the nonreciprocal reactive Cahn-Hilliard model (53) with one conservation law for inter- and intralattice spin-exchange dynamics. In each case, we have analyzed the associated linear instabilities of uniform steady states and have provided conditions under which (conserved) Hopf, Allen–Cahn, and Cahn–Hilliard-type instabilities arise, offering a direct connection between microscopic interactions and emergent spatiotemporal patterns. Finally, we have demonstrated how the three types of kinetic updates, namely single spin-flip dynamics, intralattice spin-exchange, and interlattice spin-exchange, can be combined to construct sixteen distinct nonreciprocal partial differential equation models that incorporate zero (ten models), one (five models) or two (one model) conservation laws.

In the limit of purely reciprocal interactions, we have shown that the underlying free energy functional $\mathcal{F}[m^a, m^b]$ defined in Eq. (19) serves as a Lyapunov functional for the corresponding dynamical equations. In the case of net antiferromagnetic interactions ($J_a + J_b < 0$), this free energy leads to a divergence of the linear growth rate in the ultraviolet limit ($|\mathbf{k}| \to \infty$), as observed in, for example, Fig. 2(d)–(f). To regularize this divergence and ensure linear stability at short wavelengths, we can include higher-order gradient terms in the expansion of the free energy functional Eq. (17). This yields an extended expression for the free energy:

$$\mathcal{F}[m^a, m^b] = \frac{1}{2} \int d\mathbf{x} \left[ f(m^a, m^b) + \frac{1}{12} \sum_\mu J_\mu \left( 12|\nabla m^\mu|^2 - (\partial_x^2 m^\mu)^2 - (\partial_y^2 m^\mu)^2 \right) \right],$$

where we assume vanishing Neumann boundary conditions and $f(m^a, m^b)$ is given by Eq. (20). Upon inserting this free energy into Eqs. (18) and (44), the resulting continuum equations correspond to two nonreciprocally coupled SH-type equations [25, 29] in the case of single spin-flip dynamics, or two nonreciprocally coupled PFC models [69] for intralattice spin exchange.

Building on these results, a natural next step is to move beyond linear stability analysis and investigate the fully nonlinear dynamics of each model. In particular, it would be highly valuable to study the emergence and coexistence of distinct dynamical phases—such as steady states, oscillations, and traveling patterns—using numerical bifurcation and continuation methods as applied in [18, 22, 33, 37, 69]. These techniques can uncover complex phase coexistence and bifurcations that lie beyond the reach of linear stability. Furthermore, for the nonreciprocal reactive CH model with one conservation law, it would also be insightful to apply the framework of local equilibria in diffusively coupled compartments introduced in [44, 45], offering a coarse-grained perspective on pattern formation in strongly nonlinear regimes. Moreover, going beyond the MF approximation using the pair approximation technique [81] enables the derivation of partial differential equations that capture the dynamics of the square-lattice nonreciprocal Ising model more accurately.

While the MF approximation relies on Eqs. (8) and (40), which are not strictly valid for the square lattice and in fact are known to induce an erroneous phase transition in 1D [81], the pair approximation explicitly accounts for correlation effects and does not approximate the local energetic field. Although this more accurate method has been successfully applied to describe the overall average magnetization and nearest neighbor spin correlations under single spin-flip dynamics in the nonreciprocal Ising model [14], it has yet to be extended to

such systems with spatially varying fields. Such an extension could shed light on whether a Turing-type instability is truly absent in the nonreciprocal Ising model, since it is known that the gradient-energy coefficient in the pair approximation is a nonlinear function of the coupling strength [82].

Finally, while the role of time-reversal symmetry breaking and dynamic phase transitions has been explored in the extended nonreciprocal Cahn–Hilliard model with thermal noise [34], it remains an open question how such symmetry breaking relates to dynamical phase transitions in models with zero and one conservation law.

Beyond its immediate theoretical contributions, this study provides a foundational framework for systematically deriving nonreciprocal field theories from the underlying microscopic dynamics. Although previous work has addressed the case of single-species systems with nonreciprocal interactions [3,83] and the dynamics of three-species active-passive particle mixtures [84], a comprehensive derivation of the partial differential equations for all possible combinations of conservation laws for two nonreciprocally coupled fields has remained elusive. Our results fill this gap, offering a microscopic route to macroscopic equations that is essential for understanding pattern formation in a wide range of nonequilibrium systems. In particular, our findings complement the exact hydrodynamic analysis presented in [84], which investigates nonreciprocal effective interactions in active-passive mixtures, by extending the scope to the two-field case and deriving the partial differential equations for all possible conservation laws. Furthermore, our framework permits asymmetric transition rates in position space, yielding an active spin model. In the absence of nonreciprocal couplings, a mean-field description of this class of models was developed in [85], which demonstrated the existence of a flocking transition. It would therefore be valuable to investigate how introducing a second lattice together with nonreciprocal couplings alters this behavior.

# Acknowledgements

We acknowledge valuable discussions with Daniel Greve, Lars Stutzer, and Rick Bebon.

**Funding information**    Financial support from the German Research Foundation (DFG) through the Heisenberg Program Grants GO 2762/4-1 and GO 2762/5-1 (to AG) is gratefully acknowledged.

# Software Availability

A *Mathematica* notebook containing the source code used to generate the figures, as well as a numerical solver for the differential equations, is publicly available on [86].

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
