# Peer review of "Dynamic Models for Two Nonreciprocally Coupled Fields: A Microscopic Derivation for Zero, One, and Two Conservation Laws"

_SciPost Physics_

## Round 1 · Referee Report · Anonymous (Referee 1) · 2025-10-14

Report

I thank the authors for revising the manuscript and addressing my comments. I'm very happy with the new version and recommend it for acceptance in SciPost Physics.

Just one small question regarding the author's response to my last remark, where they write
"Please note that the characteristic length scale of the SH model has nothing to do with the lattice spacing itself, as any length scale in our theory is inherently defined w.r.t. the lattice spacing. Therefore, the characteristic length scale set by the SH model is also defined w.r.t. the lattice spacing, and therefore is not the lattice spacing itself. "
That all makes sense, but if one takes the limit where the lattice spacing goes to zero, is the characteristic length scale of the SH model then defined with respect to zero (and what does this mean)?

Recommendation

Publish (surpasses expectations and criteria for this Journal; among top 10%)

  • validity: top
  • significance: top
  • originality: top
  • clarity: top
  • formatting: perfect
  • grammar: perfect

Author:  Aljaz Godec  on 2025-11-17  [id 6040]

(in reply to Report 1 on 2025-10-14)

We thank the Reviewer for this comment. To address it, we first clarify the meaning of x:

On a finite lattice, the physical (Manhattan) distance between two spins can always be written as r = k·l, where k is the number of lattice spacings (i.e., the number of spins) between the two reference spins. Importantly, k does not depend on the lattice spacing l and therefore remains well defined in the continuum limit l → 0. It simply reflects how many spins are between two reference spins.

In our manuscript, we map this discrete integer k to a continuum variable x by applying the box-averaging procedure defined on page 9. For any finite lattice, the number of spins between two reference sites is always an integer. However, after performing box averaging over sufficiently many spins, we obtain a local magnetization field defined at each point x ∈ ℝ², where x is the continuum analogue of k.

To ensure that the local magnetization m(x,t) defined on page 9 becomes a continuous real-valued function in the interval [-1,1]—and hence differentiable—we require the thermodynamic limit (N_x, N_y) → ∞. This follows from the fact that the rational numbers are dense in ℝ. Additionally, we require that the lattice spacing l → 0 so that we can apply the gradient expansion. Both limits can be taken simultaneously by considering the scaling limit (N_x, N_y) → ∞ while keeping the physical system size (L_x, L_y) = (l·N_x, l·N_y) fixed, and therefore l → 0.

With this in mind, the characteristic length scale that appears in the linear stability analysis (and therefore also in the SH model) can be interpreted as the number of spins over which the local alignment persists in the most unstable mode—that is, the number of spins over which the local magnetization is “correlated” in that mode. When the largest unstable mode appears at k=0 that means an infinite amount of spins are aligned, and when k->\Infinity it implies that no spins are aligned. The latter however can only occur in our continuum field theory (and not on a discrete lattice), since we have assumed that x is continuous and therefore patterns can emerge on arbitrarily fine scales. Taking into account higher order gradient terms in the gradient expansion would result in a regularization of this divergence for the largest unstable mode at k->\Infinity, and regularize it to a finite k value.

To conclude: x should be understood as the (continuum) counterpart of the number of spins measured from the origin x = 0, and this is well-defined even when l → 0 . It is continuous (and not discrete) because of the box-averaging procedure introduced on page 9. Under this interpretation, the characteristic length scale obtained from the linear stability analysis corresponds to the number of aligned spins associated with the most unstable mode.

---

## Round 1 · Referee Report · Lorenzo Caprini (Referee 2) · 2025-11-13

Report

In the new version of the paper, the authors have addressed my points and doubts, answering my questions.

As a consequence, I consider the paper suitable for publication in SciPost in its current form.

Recommendation

Publish (surpasses expectations and criteria for this Journal; among top 10%)

---

## Round 1 · Author Response

The replies to the specific reviewers' comments are given in the dedicated replies and in the attached rebuttal letter.

---

## Round 1 · List of Changes

List of changes

In the revised version (attached) we have:

  • Improved the grammar, corrected typos, and clarified and reformulated several sentences throughout the manuscript.
  • Highlighted the differences between our work and that in references [14] and [26].
  • Better explained the model parameters in section 2.1.
  • Added section 2.3 to highlight the significance of the nonreciprocal Ising model.
  • Added an explanation of the gradient expansion in section 3.3.
  • Added sections 3.4 and 5.4 to show how the partial differential equations are expanded close to the stationary solution.
  • Completed the missing models in table I.
  • Highlighted in the conclusion the shortcoming of the MF approximation, and how one can go beyond the MF approximation.

---

## Round 2 · Author Response

We corrected some minor remaining typos on pages 25 and 26.

---

## Editorial Decision

accepted_in_target_journal